Corrected: Author correction

# Growth hormone regulates neuroendocrine responses to weight loss via AgRP neurons

Isadora C. Furigo [1], Pryscila D.S. Teixeira[1], Gabriel O. de Souza[1], Gisele C.L. Couto[1], Guadalupe García Romero[1,2], Mario Perelló[2], Renata Frazão[3], Lucila L. Elias[4], Martin Metzger[1], Edward O. List[5], John J. Kopchick[5] & J. Donato Jr [1]

Weight loss triggers important metabolic responses to conserve energy, especially via the fall in leptin levels. Consequently, weight loss becomes increasingly difficult with weight regain commonly occurring in most dieters. Here we show that central growth hormone (GH) signaling also promotes neuroendocrine adaptations during food deprivation. GH activates agouti-related protein (AgRP) neurons and GH receptor (GHR) ablation in AgRP cells mitigates highly characteristic hypothalamic and metabolic adaptations induced by weight loss. Thus, the capacity of mice carrying an AgRP-specific GHR ablation to save energy during food deprivation is impaired, leading to increased fat loss. Additionally, administration of a clinically available GHR antagonist (pegvisomant) attenuates the fall of whole-body energy expenditure of food-deprived mice, similarly as seen by leptin treatment. Our findings indicate GH as a starvation signal that alerts the brain about energy deficiency, triggering key adaptive responses to conserve limited fuel stores.

[1] Department of Physiology and Biophysics, Institute of Biomedical Sciences, University of São Paulo, Av. Prof. Lineu Prestes, 1524, São Paulo, SP 05508-000, Brazil. [2] Laboratory of Neurophysiology, Multidisciplinary Institute of Cell Biology, Calle 526 y Camino General Belgrano, La Plata, BA 1900, Argentina. [3] Department of Anatomy, Institute of Biomedical Sciences, University of São Paulo, Av. Prof. Lineu Prestes, 2415, São Paulo, SP 05508-900, Brazil. [4] Department of Physiology, School of Medicine of Ribeirão Preto, University of São Paulo, Av. Bandeirantes, 3900, Ribeirão Preto, SP 14049-900, Brazil. [5] Edison Biotechnology Institute and Heritage College of Osteopathic Medicine, Ohio University, Konneker Research Center 206A, Athens, OH 45701, USA. Correspondence and requests for materials should be addressed to J.D.Jr (email: jdonato@icb.usp.br)

**S**everal energy-saving adaptations are triggered by the hypothalamus during food deprivation, including increases in skeletal muscle work efficiency, and inhibition of thermogenesis, thyroid and reproductive axes[1–7]. The fall in leptin levels is a starvation signal that plays a critical role inducing endocrine and autonomic adaptations during situations of negative energy balance[1–4]. Accordingly, the prevention of declining leptin levels via exogenous leptin treatment attenuates starvation-induced suppression of gonadal and thyroid axes in mice and humans[1, 3]. In addition, leptin administration reverses the effects of sustained weight reduction on energy expenditure[2, 4]. However, leptin replacement does not completely prevent the neuroendocrine adaptations induced by weight loss[1, 3, 5], indicating the existence of critical additional, but still unknown, starvation signals. The identification of other cues that induce such adaptive responses is imperative since the long-term efficacy of obesity treatments is low, in part due to body's defense mechanisms that decrease energy expenditure during weight loss[7].

In the present study, we investigated the central effects of growth hormone (GH) on energy homeostasis as GH fulfills

several requisites of an energy-deficiency signal to the brain. For example, GH secretion increases during situations of nutrient deficiency, such as hypoglycemia[8] or food deprivation[3, 8–10]. Additionally, GH receptor (GHR) is widely expressed in hypothalamic areas implicated in energy balance regulation, including the arcuate nucleus (ARH)[11]. However, the functional role of central GH signaling for energy homeostasis has not been fully defined. Here, we uncovered the importance of brain GH signaling for the regulation of energy homeostasis under normal conditions and during food deprivation. Our findings indicate that although GH does not play an important role modulating the energy balance in ad libitum fed animals, GH is a key cue that signals energy deficiency to the brain, triggering neuroendocrine responses to conserve body energy stores.

## Results

**GH activates AgRP neurons to produce orexigenic responses.** To identify GH response neurons, C57BL/6 mice received intraperitoneal (i.p.) injection of either phosphate-buffered saline (PBS) or GH and their brains were processed to detect the

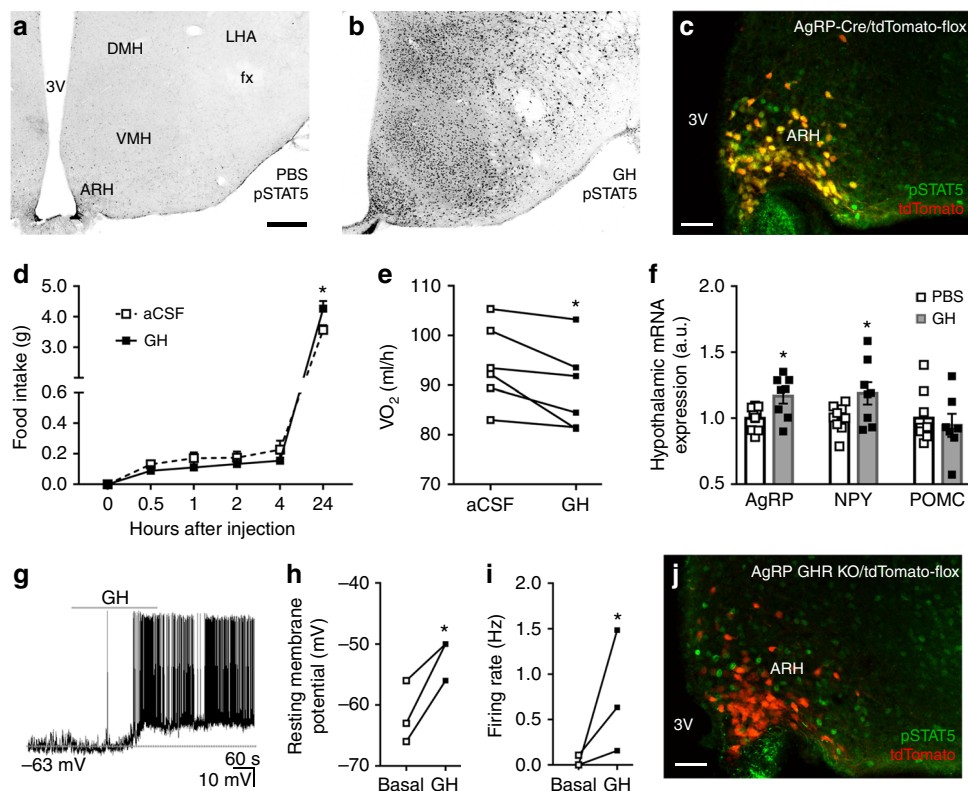

**Fig. 1** Orexigenic effect of growth hormone (GH) via activation of agouti-related protein (AgRP) neurons. **a**, **b** Photomicrographs showing the hypothalamic distribution of signal transducer and activator of transcription 5 (STAT5) phosphorylation (pSTAT5) 90 min after an intraperitoneal (i.p.) injection of phosphate-buffered saline (PBS) or porcine GH (20 μg/g body weight (b.w.)). 3V third ventricle, ARH arcuate nucleus, DMH dorsomedial nucleus, fx fornix, LHA lateral hypothalamic area, VMH ventromedial nucleus. Scale Bar = 200 μm. **c** More than 90% of AgRP neurons (red) in the ARH are responsive to porcine GH as indicated by the co-expression of pSTAT5 (green). Yellow represents double-labeled cells. Scale Bar = 50 μm.
**d** Intracerebroventricular (i.c.v.) infusion of porcine GH (6 μg in 2 μL) increased food intake (0.5 h: $t_{(8)} = 1.258$, $P = 0.244$; 1 h: $t_{(8)} = 2.075$, $P = 0.0717$; 2 h: $t_{(8)} = 1.425$, $P = 0.1919$; 4 h: $t_{(8)} = 1.518$, $P = 0.1675$; 24 h: $t_{(8)} = 2.801$, $P = 0.0232$; $n = 9$), compared to the infusion of artificial cerebrospinal fluid (aCSF). **e** The i.c.v. infusion of porcine GH reduced energy expenditure ($t_{(5)} = 3.193$, $P = 0.0242$, $n = 6$; paired $t$-test) of C57BL/6 mice. **f** Hypothalamic gene expression in C57BL/6 mice that received i.p. infusion of either PBS or porcine GH (AgRP: $t_{(15)} = 2.723$, $P = 0.0157$; neuropeptide Y (NPY): $t_{(15)} = 2.144$, $P = 0.0488$; proopiomelanocortin (POMC): $t_{(14)} = 0.5188$, $P = 0.612$; $n = 9$; unpaired $t$-test). **g** Representative whole-cell patch-clamp recording of a GH responsive AgRP neuron. Dashed line indicates the resting membrane potential. Porcine GH (5 μg/mL) was applied to the bath for approximately 5 min.
**h**, **i** Increased resting membrane potential ($t_{(2)} = 4.768$, $P = 0.0413$; paired $t$-test) and firing rate ($t_{(2)} = 3.001$, $P = 0.0477$; paired $t$-test) of GH-responsive AgRP neurons ($n = 3$). **j** AgRP growth hormone receptor knockout (GHR KO) mice showed very few GH-induced pSTAT5 (green) in AgRP neurons (red). Scale Bar = 50 μm. All results were expressed as mean ± s.e.m. *$P < 0.05$

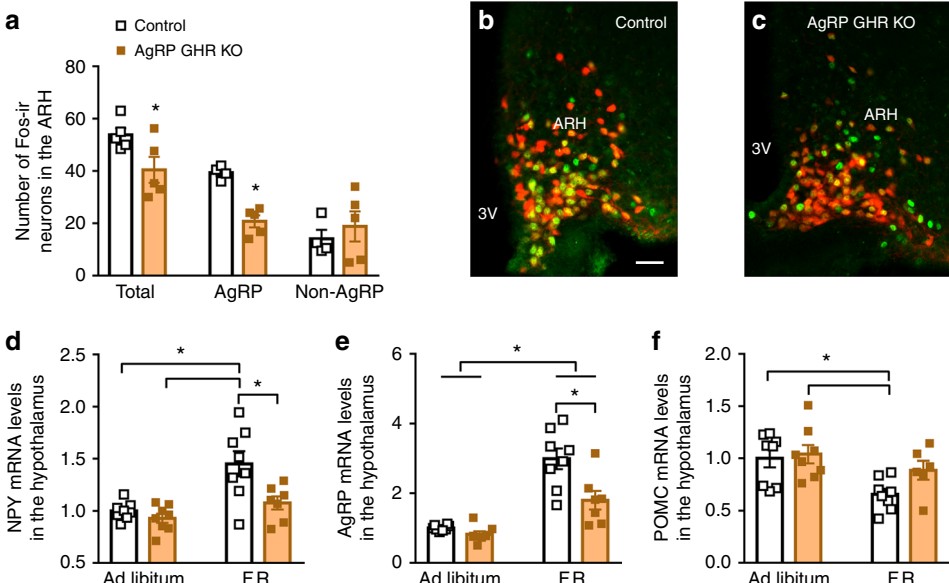

**Fig. 2** Hypothalamic changes induced by weight loss are attenuated in agouti-related protein (AgRP) growth hormone receptor knockout (GHR KO) mice. **a** AgRP GHR KO mice show reduced number of c-Fos-positive cells after 24 h of fasting in the arcuate nucleus (ARH) ($t_{(8)} = 2.348$, $P = 0.0443$, $n = 5$) and in AgRP neurons ($t_{(7)} = 6.62$, $P = 0.0003$), but not in non-AgRP cells ($t_{(7)} = 0.6529$, $P = 0.5347$). **b**, **c** Representative photomicrographs showing fasting-induced c-Fos expression (green) and the co-localization with AgRP neurons (red). Scale Bar = 50 µm. **d** Hypothalamic mRNA expression of neuropeptide Y (NPY) (main effect of food restriction (F.R.) [$F_{(1, 27)} = 16.44$, $P = 0.0004$], main effect of GHR ablation [$F_{(1, 27)} = 9.036$, $P = 0.0057$] and interaction [$F_{(1, 27)} = 4.215$, $P = 0.0499$]; $n = 7$–8). **e** Hypothalamic mRNA expression of AgRP (main effect of F.R. [$F_{(1, 27)} = 54.67$, $P < 0.0001$], main effect of GHR ablation [$F_{(1, 27)} = 11.64$, $P = 0.002$] and interaction [$F_{(1, 27)} = 6.417$, $P = 0.0174$]; $n = 7$–8). **f** Hypothalamic mRNA expression of proopiomelanocortin (POMC) (main effect of F.R. [$F_{(1, 26)} = 9.582$, $P = 0.0047$], main effect of GHR ablation [$F_{(1, 26)} = 2.813$, $P = 0.1055$] and interaction [$F_{(1, 26)} = 1.399$, $P = 0.2476$]; $n = 6$–8). The effects of F.R. were analyzed by two-way analysis of variance (ANOVA). All results were expressed as mean ± s.e.m. *$P < 0.05$

phosphorylation of signal transducer and activator of transcription 5 (pSTAT5), a marker of GHR activation[11]. We observed that GH robustly induced pSTAT5 in several hypothalamic nuclei, whereas few pSTAT5-positive cells were found in PBS-injected mice (Fig. 1a, b). Since agouti-related protein (AgRP) neurons in the ARH are major regulators of energy homeostasis[12], we investigated whether they are responsive to GH. We found that 91.2 ± 3.0% of AgRP neurons presented pSTAT5 after i.p. GH injection (Fig. 1c), suggesting that GH acts on AgRP cells. Although intracerebroventricular (i.c.v.) administration of GH caused no significant changes in food intake during the first 4 h of measurement, C57BL/6 mice exhibited increased food intake and reduced energy expenditure 24 h after the injection (Fig. 1d, e). GH injection also increased hypothalamic AgRP and neuropeptide Y (NPY) messenger RNA (mRNA) levels, whereas proopiomelanocortin (POMC) expression remained unaffected (Fig. 1f). Thus, GH injection mimicked the effects induced by chemogenetic activation of AgRP neurons[13]. To confirm that GH activates AgRP neurons, whole-cell patch-clamp recordings were performed in brain slices of AgRP-reporter mouse (Supplementary Fig. 1). We found that 25% of ARH AgRP neurons (3 out 12 recorded cells from 5 mice) were depolarized by GH (Fig. 1g), increasing the resting membrane potential and action potential frequency of responsive cells, compared to baseline (Fig. 1h, i). In order to determine whether the effect of GH is direct in AgRP cells (independent of action potential-mediated synaptic transmission), a new set of recordings was performed in the presence of the voltage-gated sodium channel antagonist tetrodotoxin (TTX; 1 µM) and synaptic blockers (20 µM 6-cyano-7-nitroquinoxaline-2,3-dione (CNQX), 50 µM 2-amino-5-phosphonovalerate (Ap-5) and 50 µM picrotoxin). We also found that GH application in the presence of TTX and synaptic blockers depolarized 25% of ARH AgRP neurons (3 out 12 recorded cells from 4 mice), changing in +7.7 ± 1.4 mV their

resting membrane potential ($t_{(2)} = 5.277$, $P = 0.0341$). Altogether, these findings indicate that exogenous administration of GH induces an orexigenic response via activation of AgRP neurons.

**GHR ablation in AgRP neurons causes no metabolic imbalances.** To study in detail the importance of GH signaling in AgRP neurons, we generated mice carrying an AgRP-specific GHR ablation. As expected, AgRP GHR knockout (KO) mice did not show GH-induced pSTAT5 in AgRP neurons (Fig. 1j), while a normal pSTAT5 distribution was observed in other neuronal populations (Supplementary Fig. 2). AgRP GHR KO mice displayed a similar body weight, food intake, energy expenditure, respiratory quotient, ambulatory activity, adiposity, lean body mass, body length, glucose tolerance and insulin sensitivity compared to control animals (Supplementary Fig. 3a-h and Supplementary Fig. 4a, b). GHR ablation in AgRP cells also did not affect leptin sensitivity (Supplementary Fig. 5), ghrelin-induced food intake or ghrelin-induced c-Fos expression in the ARH (Supplementary Fig. 6a–d). These results suggest that GHR expression in AgRP neurons is unnecessary for the regulation of energy homeostasis under normal circumstances or for the response to key hormones that rely on AgRP neurons to modulate energy homeostasis. Thus, endogenous fluctuations of plasma GH levels likely do not play an important role modulating the energy balance in ad libitum fed mice.

**GH triggers neuroendocrine adaptations during weight loss.** AgRP neurons express c-Fos during food deprivation as an indicator of increased cell activity[14, 15]. Notably, the amount of fasting-induced c-Fos-positive cells in the ARH was reduced in AgRP GHR KO mice (Fig. 2a–c), despite the number of ARH AgRP cells remained unchanged (Supplementary Fig. 7). We then introduced an AgRP-tdTomato genetic background to our mouse

model to study if the ablation of GHR in AgRP cells impairs fasting-induced increments of c-Fos in AgRP neurons. We found that the number AgRP neurons positive for c-Fos was smaller in fasted AgRP GHR KO mice as compared to fasted control mice, while the number of non-AgRP cells positive for c-Fos remained unchanged between the groups (Fig. 2a–c). This observation suggests that AgRP neurons are unable to appropriately sense food deprivation without GH signaling. Since AgRP neurons are critically involved in neuroendocrine adaptations induced by weight loss[12, 16, 17], control and AgRP GHR KO mice were further studied either in ad libitum or during 60% food restriction (F.R.). Two days of F.R. (40% of the normal intake) was sufficient to increase hypothalamic NPY and AgRP expression in control mice, whereas GHR ablation in AgRP neurons prevented these effects (Fig. 2d, e). F.R. reduced hypothalamic POMC expression, but AgRP-specific GHR ablation did not significantly affect this response (Fig. 2f). Next, we assessed whether AgRP GHR KO mice exhibit a normal endocrine response to weight loss. F.R. reduced serum T4 and testosterone concentrations in control mice, whereas AgRP GHR KO mice showed a blunted suppression of these hormones (Fig. 3a, b). Additionally, F.R. increased serum corticosterone concentration in control mice, but this increase was prevented in AgRP GHR KO mice (Fig. 3c). While F. R. also reduced serum leptin levels (Fig. 3d) and increased serum GH concentration (Fig. 3e), GHR ablation in AgRP cells caused no significant effects in these responses. F.R. or GHR ablation in AgRP cells did not affect circulating prolactin levels (Supplementary Fig. 8). We also assessed interscapular brown adipose tissue uncoupling protein-1 (UCP-1) mRNA. F.R. suppressed UCP-1 expression in control mice, whereas this reduction was attenuated in AgRP GHR KO animals (Fig. 3f). Altogether, our findings strongly indicate that GH signaling in AgRP neurons is required for the induction of key neuroendocrine responses that conserve energy during F.R.

**Energy-conserving effects of GH signaling in AgRP neurons**. To test if the lack of adaptive responses to energy deficits significant impacts on energy balance, we recorded the whole-body energy expenditure of both groups of mice during 60% F.R. Control mice decreased their energy expenditure during F.R. (Fig. 4a and Supplementary Fig. 9), which is in accordance with the adaptive responses that conserve energy during this situation[7]. However, the decrease in energy expenditure of AgRP GHR KO mice during F.R. was significantly smaller, as compared to control mice, suggesting that they did not save energy as efficiently as control mice (Fig. 4a and Supplementary Fig. 9). Consequently, AgRP GHR KO mice exhibited a higher rate of weight loss (Fig. 4b), which was predominantly due to fat mass loss (Fig. 4c), but also to lean mass loss (Fig. 4d). No differences between groups were observed in the respiratory quotient or ambulatory activity during F.R. (Supplementary Fig. 10). Since GH secretion during F.R. is essential to preserve blood glucose[9, 10], we daily monitored glycemia and found that AgRP GHR KO mice exhibited lower glycemia during the initial days of F.R. when compared to control animals (Fig. 4e).

As GH secretion displays sexual dimorphism[18–20], we also determined whether AgRP GHR KO females show blunted neuroendocrine responses to F.R. As seen in males, AgRP GHR KO females showed an attenuated increase in hypothalamic AgRP and NPY mRNA levels during F.R., whereas POMC gene expression was not affected by GHR ablation (Supplementary Fig. 11a–c). In contrast to our observations in male mice, the decrease in energy expenditure of AgRP GHR KO females during F.R. was similar as seen in control females (Supplementary Fig. 11d). However, AgRP GHR KO females exhibited increased

weight loss (Supplementary Fig. 11e) and decreased glycemia (Supplementary Fig. 11f) during F.R., as compared to control females, which is similar to our observations in control vs. AgRP GHR KO males.

To test whether GH signaling in AgRP neurons regulates energy homeostasis in other conditions of nutrient deficiency, control and AgRP GHR KO male mice received i.p. injection of 2-deoxi-D-glucose (2DG), which causes glucoprivation[21]. The counter-regulatory response triggered by 2DG was similar between the groups (Supplementary Fig. 12). However, control mice exhibited the expected 2DG-induced hyperphagia[22], while this response was prevented in AgRP GHR KO mice (Fig. 4f).

**GHR ablation in leptin receptor cells or the entire brain**. Since distinct leptin receptor (LepR)-expressing neurons are known to be involved in the metabolic adaptations that conserve energy during F.R.[6, 13, 16, 17, 23], we generated mice carrying GHR ablation either in LepR cells (Fig. 5a–c and Supplementary Fig. 13) or the entire brain (nestin-derived cells; Fig. 5d and Supplementary Fig. 14). Thus, we investigated whether GHR ablation in a large number of neuronal populations produces additional effects on the metabolic adaptations induced by weight loss. In contrast to AgRP GHR KO mice, both LepR GHR KO and brain GHR KO mice exhibited increased body weight and length (Fig. 5e, f). However, LepR GHR KO mice showed reduced body adiposity and serum leptin levels compared to control animals, whereas brain GHR KO mice had higher lean body mass and upregulation of hypothalamic GH-releasing hormone expression (GHRH; Fig. 5g–j). Thus, central GHR ablation likely impaired GH negative feedback, leading to increased body growth, especially in brain GHR KO mice. Despite these changes, food intake, energy expenditure, respiratory quotient, ambulatory activity and leptin sensitivity were not affected in these mice (Supplementary Fig. 15a–e).

Next, we investigated whether LepR GHR KO and brain GHR KO mice were able to reduce their energy expenditure during F.R. As seen in AgRP GHR KO mice, both LepR GHR KO and brain GHR KO mice showed defects in their ability to save energy during F.R. (Fig. 6a), leading to a greater weight loss (Fig. 6b). Notably, a sharp decline in energy expenditure of LepR GHR KO mice was observed during the last days of F.R. (Fig. 6a), which may reflect an early depletion of their body energy reserves since these mice had lower adiposity (Fig. 5g). Hence, some LepR GHR KO mice became lethargic during F.R. and had to be killed and removed from the study (Supplementary Fig. 16). As seen in AgRP GHR KO mice, GHR ablation in LepR-expressing cells prevented the F.R.-induced suppression of UCP-1 expression in the brown adipose tissue (BAT; Fig. 6c). LepR GHR KO mice also exhibited lower glycemia during F.R. (Fig. 6d) and attenuated 2DG-induced hyperphagia compared to control or brain GHR KO mice (Fig. 6e). Thus, we confirmed the ability of central GH signaling to promote energy saving adaptations during F.R. in different mouse models; however, GHR ablation in broader neural populations did not produce additional effects as compared to AgRP GHR KO mice, emphasizing a crucial role of AgRP neurons mediating GH responses.

**Pegvisomant increases the metabolic rate of food-deprived mice**. To assess whether the energy-conserving effects of GH can be pharmacologically manipulated, we tested in mice the effects of a clinically available GHR antagonist (pegvisomant). Initially, we determined whether pegvisomant treatment inhibits GH signaling in mice. We found that C57BL/6 mice treated with two daily i.p. injections of pegvisomant during four consecutive days showed reduced plasma insulin-like growth factor-1 (IGF-1)

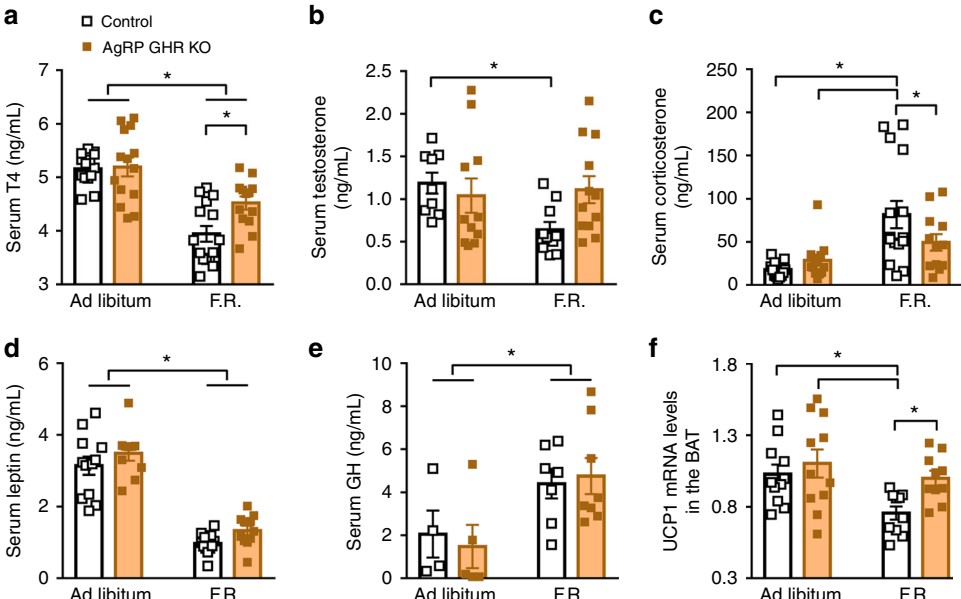

**Fig. 3** Neuroendocrine changes induced by weight loss are attenuated in agouti-related protein (AgRP) growth hormone receptor knockout (GHR KO) mice. **a** Serum concentration of T4 (main effect of food restriction (F.R.) [$F_{(1, 53)} = 46.18$, $P < 0.0001$], main effect of GHR ablation [$F_{(1, 53)} = 2.796$, $P = 0.1004$] and interaction [$F_{(1, 53)} = 4.953$, $P = 0.0303$]). **b** Serum concentration of testosterone (main effect of F.R. [$F_{(1, 38)} = 2.36$, $P = 0.1327$], main effect of GHR ablation [$F_{(1, 38)} = 1.086$, $P = 0.3039$] and interaction [$F_{(1, 38)} = 3.949$, $P = 0.0541$]). **c** Serum concentration of corticosterone (main effect of F.R. [$F_{(1, 49)} = 16.13$, $P = 0.0002$], main effect of GHR ablation [$F_{(1, 49)} = 1.072$, $P = 0.3055$] and interaction [$F_{(1, 49)} = 4.13$, $P = 0.0476$]). **d** Serum concentration of leptin (main effect of F.R. [$F_{(1, 41)} = 140.3$, $P < 0.0001$], main effect of GHR ablation [$F_{(1, 41)} = 3.803$, $P = 0.058$] and interaction [$F_{(1, 41)} = 0.0023$, $P = 0.9617$]). **e** Serum concentration of growth hormone (GH (main effect of F.R. [$F_{(1, 20)} = 9.486$, $P = 0.0059$], main effect of GHR ablation [$F_{(1, 20)} = 0.0157$, $P = 0.9013$] and interaction [$F_{(1, 20)} = 0.2558$, $P = 0.6186$]). **f** UCP-1 mRNA expression in the interscapular brown adipose tissue (BAT; main effect of F.R. [$F_{(1, 38)} = 7.158$, $P = 0.0109$], main effect of GHR ablation [$F_{(1, 38)} = 5.196$, $P = 0.0283$] and interaction [$F_{(1, 38)} = 1.449$, $P = 0.2361$]; $n = 11-15$). The data were analyzed by two-way analysis of variance (ANOVA). All results were expressed as mean ± s.e.m. *$P < 0.05$

levels (Fig. 7a) and higher GHRH mRNA in the hypothalamus (Fig. 7b), as compared to PBS-treated mice. These results show that pegvisomant impairs GH signaling; thus, this drug acts as a GHR antagonist in mice. Next, C57BL/6 mice were subjected to 60% F.R. while receiving two daily i.p. injections of either pegvisomant or mouse recombinant leptin. Pegvisomant attenuated the decline in energy expenditure during F.R. to the same magnitude than leptin (Fig. 7c). However, this prevention was observed only in the second day of F.R. in both groups and did not affect significantly the weight loss (Fig. 7c, d). In addition, pegvisomant treatment did not change the glycemia during F.R., whereas leptin-treated mice showed a reduction in glycemia, especially during the last days of F.R. (Fig. 7e).

## Discussion

Clinical trials have indicated that leptin administration failed as an efficient therapeutic approach to treat obesity[24–26], although it attenuates the neuroendocrine and metabolic changes induced by weight loss[1–5]. Actually, many researchers agree that leptin's main role is to signal starvation (via falling leptin levels) and consequently produce appropriate behavioral and metabolic responses that increase the chances of survival[27, 28]. Our findings point out that GH is an additional cue that signals energy deficiency to the brain, triggering key adaptive responses to conserve body energy stores (Fig. 7f). These findings help to explain why leptin replacement does not completely reverse the neuroendocrine adaptations induced by weight loss[1, 3, 5], since both GH and leptin play a role informing the brain about energy deficiency.

Our study also presents further evidence that AgRP neurons are fundamental for integrating information of various starvation signals to modulate energy homeostasis[12]. Current results not only show that the activity of AgRP neurons is influenced by plasma GH levels but also highlight the brain as a key target of GH signaling.

We observed that ~90% of AgRP cells increased pSTAT5 levels in response to a systemic injection of GH. However, just 25% of ARH AgRP neurons depolarized after GH stimulus in an ex vivo setting. Methodological or biological factors may explain this quantitative difference. For instance, technical procedures inherent to the experimental approach, such as brain slicing, may impair the ex vivo responsiveness of AgRP neurons. Also, it is possible that the signaling pathway or electrical effects of GH on AgRP neurons may differ. In line with this possibility, it has been shown that only a percentage of LepR-expressing neurons, including POMC and AgRP neurons, change their resting membrane potential after leptin application[29, 30]. Our findings also suggest that the genomic actions of GH, presumably via STAT5 or STAT5-associated transcription factors, may have a more relevant physiological role than its effects on resting membrane potential. Indeed, knockout mice for *Stat5a/b* genes exhibit similar defects as those caused by GH deficiency, at least in terms of body and tissue growth[31–33].

Here, c-Fos expression in ARH AgRP neurons was measured after 24 h of fasting, while the importance of endogenous GH signaling in AgRP cells for the neuroendocrine responses to weight loss was evaluated in calorie-restricted mice that received 40% of their normal intake. Such F.R. protocol was chosen for the current study because it was previously shown to induce robust increments of plasma GH levels[9, 10]. Furthermore, 2 days of F.R. produced similar neuroendocrine effects in mice than those caused by an acute fast, including suppression of thyroid and reproductive axes, increased glucocorticoid secretion, increase in hypothalamic AgRP and NPY expression and decreased POMC

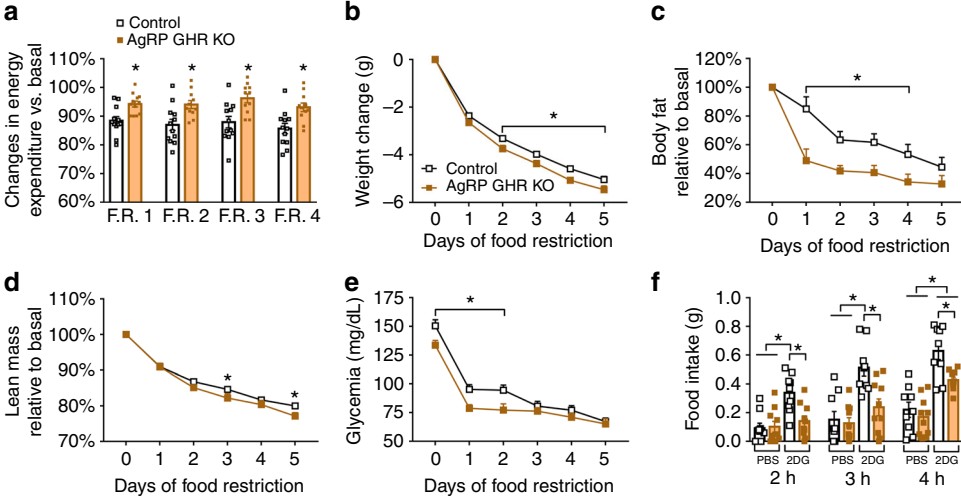

**Fig. 4** Energy-conserving effects of growth hormone (GH) signaling in agouti-related protein (AgRP) neurons. **a** Reduction in energy expenditure ($VO_2$) during food restriction (F.R.) compared to baseline (F.R. 1, $t_{(22)} = 3.296$, $P = 0.0033$; F.R. 2, $t_{(21)} = 2.913$, $P = 0.0083$; F.R. 3, $t_{(21)} = 3.223$, $P = 0.0041$; F.R. 4, $t_{(21)} = 3.144$, $P = 0.0049$; $n = 11$–12; unpaired $t$-test). **b** Changes in body weight (main effect of F.R. [$F_{(5, 225)} = 1525$, $P < 0.0001$], main effect of growth hormone receptor (GHR ablation [$F_{(1, 45)} = 7.077$, $P = 0.0108$] and interaction [$F_{(5, 225)} = 3.251$, $P = 0.0074$]; control = 25; AgRP GHR KO = 22). **c** Body fat mass (main effect of F.R. [$F_{(5, 60)} = 29.23$, $P < 0.0001$], main effect of GHR ablation [$F_{(1, 12)} = 11.27$, $P = 0.0057$] and interaction [$F_{(5, 60)} = 2.079$, $P = 0.0805$]; control = 5; AgRP GHR KO = 9). **d** Lean body mass (main effect of F.R. [$F_{(5, 60)} = 526.9$, $P < 0.0001$], main effect of GHR ablation [$F_{(1, 12)} = 3.429$, $P = 0.0888$] and interaction [$F_{(5, 60)} = 3.026$, $P = 0.0168$]). **e** Blood glucose changes during F.R. (main effect of time [$F_{(5, 158)} = 88.32$, $P < 0.0001$], main effect of GHR ablation [$F_{(1, 158)} = 19.45$, $P < 0.0001$] and interaction [$F_{(5, 158)} = 1.463$, $P = 0.205$]; $n = 14$–15) and Fisher's least significant difference (LSD) post-hoc test (*$P < 0.01$). **f** Food intake after intraperitoneal (i.p.) injection of either phosphate-buffered saline (PBS) or 2-deoxy-D-glucose (2DG; 0.5 mg/kg body weight (b.w.); $n = 9$–10). The effects of F.R. or 2DG were analyzed by two-way analysis of variance (ANOVA). All results were expressed as mean ± s.e.m.

expression[1, 15]. Additionally, AgRP GHR KO mice showed no change in serum leptin levels compared to control animal, while they lost more fat mass during F.R. Although the reasons for the lack of difference in circulating leptin levels in animals with different degrees of adiposity are unknown, it is possible to speculate that changes in autonomic innervation or hormonal milieu could have affected leptin synthesis and secretion by adipose tissue. In fact, POMC and AgRP neurons modulate autonomic nerve activity in various tissues, including fat depots[34], and these neurons can regulate fasting-induced fall in leptin levels, independently of changes in fat mass[35].

Some differences were observed between our conditional knockout models. While AgRP GHR KO mice showed no changes in growth or basal metabolism, both LepR GHR KO and brain GHR KO mice displayed increased weight gain over time, associated with a higher body length. The lack of difference in the growth of AgRP GHR KO mice, as compared to control mice, is interesting since previous studies suggested that NPY plays a role regulating GH secretion and consequently somatic growth[36–38]. Thus, other populations of NPY neurons may be involved with this control or simply GHR expression does not affect how ARH AgRP/NPY neurons modulate the somatotropic axis. On the other hand, GHR ablation in the entire brain was expected to increase GH secretion due to impaired GH negative feedback. Indeed, we observed a significant increase in GHRH mRNA levels in the hypothalamus of brain GHR KO mice. A recent study showed that ~45% of ARH GHRH-positive neurons are responsive to leptin[39]. Thus, GHR ablation in LepR cells probably increased body length by affecting GH negative feedback in a subpopulation of ARH GHRH neurons. LepR GHR KO mice also showed decreased adiposity, although no differences in food intake and energy expenditure were observed in this model. Since the reason for the lower body fat mass in LepR GHR KO mice is unknown, we believe that it is likely related to a larger number of cells affected by GHR deletion, including peripheral cells that also express the LepR. Although the energy-conserving effects of GH signaling were clearly confirmed in both LepR GHR KO and brain GHR KO mice, only LepR GHR KO animals showed reduced glycemia during F.R. and a blunted 2DG-induced hyperphagia. It is possible that genetic deletions affecting a large number of cells may produce more pronounced compensatory effects during development masking the effects induced by the lack of GHR in the entire brain. In addition, Nestin-Cre transgene expression was previously shown to produce a phenotype per se which could have interfered with the responses observed in the present study[40, 41].

Pegvisomant is a modified version of human GH that has a much higher half-life than GH and acts as a competitive antagonist of the GHR[42, 43]. Here, we show that daily treatment with pegvisomant in mice reduces plasma IGF-1 levels and increases hypothalamic GHRH mRNA expression, strongly suggesting that this compound impairs GH signaling. Importantly, our subsequent study with pegvisomant treatment provided a proof of concept that a clinically available GHR antagonist produces energy-saving adaptations during food deprivation. Thus, pharmacological compounds that are capable of targeting GH signaling may prevent compensatory decreases in energy expenditure during F.R. and consequently represent a promising approach to facilitate weight loss and improve the efficacy of obesity treatments. However, additional pharmacological studies are needed to improve the treatment efficacy in terms of dose, route of administration or combination of administered drugs. In conclusion, our findings uncovered a novel physiological role for GH, which is to signal caloric deficiency to the brain, triggering important adaptive responses to conserve energy via activation of AgRP neurons.

## Methods

**Mice**. To induce genetic ablation of the GHR, mice carrying loxP-flanked *Ghr* alleles[44] were bred either with AgRP-IRES-Cre mouse (Agrp^tm1(cre)Lowl/J, The

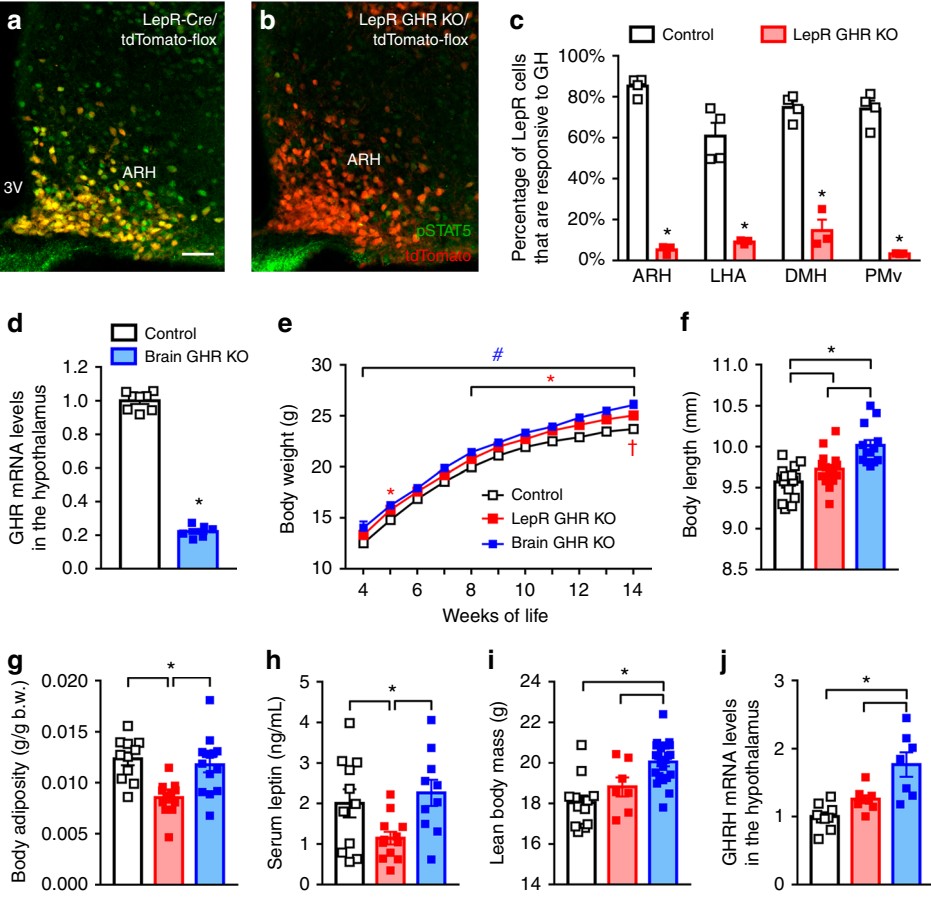

**Fig. 5** Consequences of growth hormone receptor (GHR) ablation in leptin receptor (LepR)-expressing cells or the entire brain. **a–c** A high percentage of LepR neurons (red) in the arcuate nucleus (ARH; $t_{(5)}$ = 28.42, $P$ < 0.0001), lateral hypothalamic area (LHA; $t_{(5)}$ = 6.777, $P$ = 0.0011), dorsomedial nucleus (DMH; $t_{(5)}$ = 10.51, $P$ = 0.0001) and ventral premammillary nucleus (PMv; $t_{(5)}$ = 14.6, $P$ < 0.0001) is responsive to porcine growth hormone (GH) (20 μg/g body weight (b.w.)) in control mice (phosphorylation of signal transducer and activator of transcription 5 (pSTAT5), green), whereas very few pSTAT5 is observed in tdTomato cells of LepR GHR KO mice ($n$ = 3–4; unpaired $t$-test). Yellow represents double-labeled cells. Scale Bar = 50 μm. **d** GHR mRNA expression in the hypothalamus of control and brain GHR KO mice ($t_{(12)}$ = 30.02, $P$ < 0.0001, $n$ = 6–8; unpaired $t$-test). **e** Body weight changes in control, LepR GHR KO and brain GHR KO mice (main effect of time [$F_{(10, 1036)}$ = 370.9, $P$ < 0.0001], main effect of GHR ablation [$F_{(2, 1036)}$ = 76.28, $P$ < 0.0001] and interaction [$F_{(20, 1036)}$ = 0.4301, $P$ = 0.9867]; two-way analysis of variance (ANOVA); $n$ = 14–20; *$P$ < 0.05, LepR GHR KO vs. control mice; #$P$ < 0.05, brain GHR KO vs. control mice; †$P$ < 0.05, LepR GHR KO vs. brain GHR KO mice). **f** Body length ($F_{(2, 42)}$ = 16.29, $P$ < 0.0001, $n$ = 13–17). **g** Body adiposity ($F_{(2, 34)}$ = 9.815, $P$ = 0.0004, $n$ = 11–14). **h** Serum leptin concentration ($F_{(2, 30)}$ = 4.477, $P$ = 0.0199, $n$ = 10–12/group). **i** Lean body mass ($F_{(2, 38)}$ = 13.17, $P$ < 0.0001, $n$ = 7–22/group). **j** Hypothalamic GHRH mRNA expression ($F_{(2, 21)}$ = 13.43, $P$ = 0.0001, $n$ = 7–9) of 6-month-old male mice. One-way ANOVA and the Newman–Keuls test were used when the data of control, LepR GHR KO and brain GHR KO mice were compared. All results were expressed as mean ± s.e.m.

Jackson Laboratory), LepR-IRES-Cre mouse (B6.129-Lepr^tm2(cre)Rck/J, The Jackson Laboratory) or Nestin-Cre mouse (B6.Cg-Tg^(Nes-cre)1Kln/J, The Jackson Laboratory). Control mice were homozygous for the loxP-flanked *Ghr* allele, whereas their littermates carrying the Cre alleles were considered the conditional knockout mice. In some histological and electrophysiological experiments, AgRP-IRES-Cre or LepR-IRES-Cre mice were also crossed with the Cre-inducible tdTomato-reporter mouse (B6;129S6-Gt(ROSA)26Sor^tm9(CAG-tdTomato)Hze/J, The Jackson Laboratory), allowing the visualization of AgRP or LepR neurons via expression of the tdTomato fluorescent protein. The mice in these strains were in the C57BL/6 background. Mice were weaned at 3–4 weeks of age and their mutations were confirmed by genotyping the DNA that had been previously extracted from the tail tip (REDExtract-N-Amp™ Tissue PCR Kit, Sigma). The genetically modified mouse models and wild-type C57BL/6 mice were produced and maintained in standard conditions of light (12 h light/ dark cycle) and temperature (22 ± 1 °C). Mice received a regular rodent chow diet (2.99 kcal/g; 9.4% calories from fat). All experiments were carried out in compliance with the National Institutes of Health (NIH) guidelines for the care and use of laboratory animals and were previously approved by the Ethics Committee on the Use of Animals of the Institute of Biomedical Sciences at the University of São Paulo.

**Brain histology**. To visualize GH-responsive cells in the brain, adult mice ($n$ = 3–4/group) received an acute i.p. injection of porcine pituitary GH (pGH; 20 μg/g, from Dr. A.F. Parlow, National Hormone and Peptide Program (NHPP), National

Institute of Diabetes and Digestive and Kidney Diseases) and were perfused 90 min later. To assess fasting-induced c-Fos expression in the ARH, control and AgRP GHR KO mice carrying the Cre-inducible tdTomato-reporter protein ($n$ = 5/ group) were perfused after 24 h of fasting. Ghrelin-induced c-Fos expression was measured in control and AgRP GHR KO mice ($n$ = 4–5/group) perfused 90 min following a subcutaneous injection of ghrelin (0.2 μg/g body weight (b.w.), Global Peptide, cat. no. C-et-004). For the perfusions, mice were deeply anesthetized with isoflurane and perfused transcardially with saline, followed by a 10% buffered formalin solution (150–200 mL per mouse). Brains were collected and post-fixed in the same fixative for 30–60 min and cryoprotected overnight at 4 °C in 0.1 M PBS containing 20% sucrose, pH 7.4. Brains were cut (30 μm thick sections) in the frontal plane using a freezing microtome. To label pSTAT5, brain sections were rinsed in 0.02 M potassium PBS, pH 7.4 (KPBS), followed by pretreatment in an alkaline (pH > 13) water solution containing 1% hydrogen peroxide and 1% sodium hydroxide for 20 min. After rinsing in KPBS, sections were incubated in 0.3% glycine and 0.03% lauryl sulfate for 10 min each. Next, sections were blocked in 3% normal donkey serum for 1 h, followed by incubation in anti-pSTAT5^Tyr694 primary antibody (1:1000; Cell Signaling; #9351) for 40 h. For the immunofluorescence reaction, sections were rinsed in KPBS and incubated for 90 min in AlexaFluor^488-conjugated secondary antibody (1:500, Jackson Laboratories). Sections were mounted onto gelatin-coated slides and the slides were coverslipped with Fluoromount G (Electron Microscopic Sciences, Hatfield, PA). For the immunoperoxidase staining, sections were incubated for 1 h in biotin-conjugated secondary antibody (1:1000, Jackson Laboratories) and next for 1 h with an avidin-

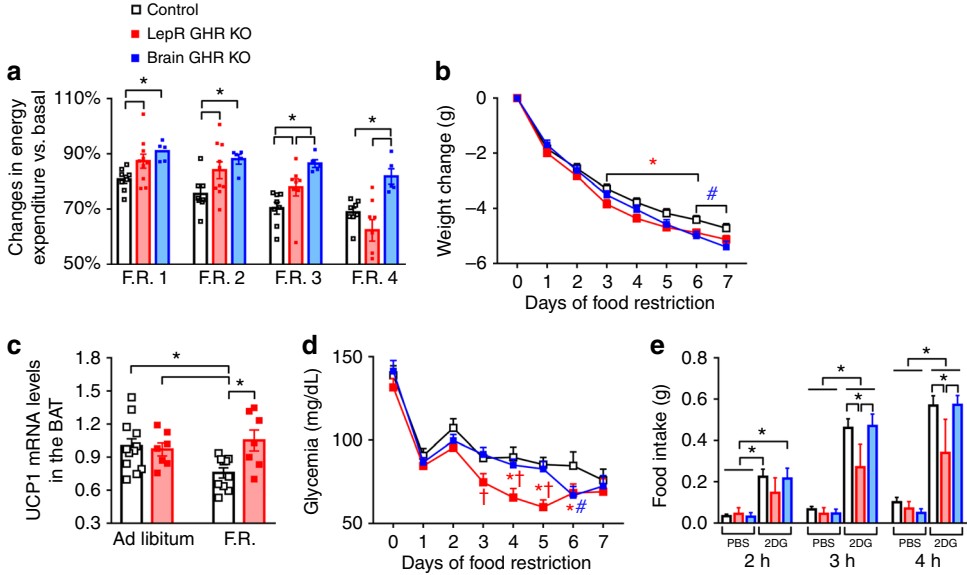

**Fig. 6** Central ablation of growth hormone receptor (GHR) prevents energy-saving adaptations during food restriction (F.R.) **a** Reduction in energy expenditure ($VO_2$) during the days of F.R. compared to baseline (F.R. 1, $F_{(2, 20)} = 4.792$, $P = 0.0199$; F.R. 2, $F_{(2, 20)} = 4.707$, $P = 0.0212$; F.R. 3, $F_{(2, 18)} = 8.695$, $P = 0.0023$; F.R. 4, $F_{(2, 18)} = 9.151$, $P = 0.0018$; $n = 5$–10). **b** Body weight changes during F.R. (main effect of F.R. [$F_{(7, 315)} = 1011$, $P < 0.0001$], main effect of GHR ablation [$F_{(2, 45)} = 2.258$, $P = 0.1163$] and interaction [$F_{(14, 315)} = 3.189$, $P = 0.0001$]; $n = 14$–20; *$P < 0.05$, LepR GHR KO vs. control mice; #$P < 0.05$, brain GHR knockout (KO) vs. control mice; Fisher's least significant difference (LSD) post-hoc test). **c** UCP-1 mRNA expression in the brown adipose tissue (BAT; main effect of F.R. [$F_{(1, 32)} = 1.374$, $P = 0.2499$], main effect of GHR ablation [$F_{(2, 32)} = 3.646$, $P = 0.0652$] and interaction [$F_{(2, 32)} = 5.547$, $P = 0.0248$]) of LepR GHR KO mice. **d** Blood glucose changes during F.R. (main effect of time [$F_{(7, 315)} = 65.76$, $P < 0.0001$], main effect of GHR ablation [$F_{(2, 45)} = 4.866$, $P = 0.0122$] and interaction [$F_{(14, 315)} = 1.491$, $P = 0.1125$]) and Fisher's LSD post-hoc test (*$P < 0.05$, LepR GHR KO vs. control mice; #$P < 0.05$, brain GHR KO vs. control mice; †$P < 0.05$, LepR GHR KO vs. brain GHR KO mice; $n = 14$–15). **e** Food intake after intraperitoneal (i.p.) injection of either phosphate-buffered saline (PBS) or 2-deoxi-D-glucose (2DG; 0.5 mg/kg body weight (b.w.); $n = 6$–14). One-way analysis of variance (ANOVA) and the Newman–Keuls test were used when the data of control, LepR GHR KO and brain GHR KO mice were compared. The changes in body weight and glycemia or the effects of 2DG were analyzed by two-way ANOVA. All results were expressed as mean ± s.e.m.

biotin complex (1:500, Vector Labs). The peroxidase reaction was performed using 0.05% 3,3'-diaminobenzidine, 0.25% nickel sulfate and 0.03% hydrogen peroxide resulting in a black nuclear staining. The slides were coverslipped with DPX mounting medium (Sigma, St. Louis, MO). The reaction to label c-Fos was similar to the pSTAT5 protocol, except that brain sections were incubated in anti-c-Fos antibody (1:20,000, Ab5, Millipore) for 48 h. Photomicrographs were acquired with a Zeiss Axiocam HRc camera coupled to a Zeiss Axioimager A1 microscope (Zeiss, Munich, Germany). Images were digitized using Axiovision software (Zeiss). The ImageJ Cell Counter software (http://rsb.info.nih.gov/ij/) was used to manually count the number of cells in the areas of interest.

**Electrophysiology.** To examine the acute effects of GH on the membrane excitability of AgRP neurons, whole-cell patch-clamp recordings were performed in hypothalamic slices of male AgRP-reporter mouse (8–12 weeks old). Mice were decapitated, their brains were collected and immediately submerged in ice-cold, carbogen-saturated (95% $O_2$ and 5% $CO_2$) artificial cerebrospinal fluid (aCSF; 124 mM NaCl, 2.8 mM KCl, 26 mM NaHCO₃, 1.25 mM NaH₂PO₄, 1.2 mM MgSO₄, 5 mM glucose and 2.5 mM CaCl₂). Coronal sections (250 μM thick) from a hypothalamic block were cut with a Leica VT1000S vibratome and then incubated in oxygenated aCSF at room temperature for at least 1 h before recording. Slices were transferred to the recording chamber and allowed to equilibrate for 10–20 min before recording. The slices were bathed in oxygenated aCSF (30 °C) at a flow rate of 2 mL/min. In current-clamp mode, current injection (<± 8 pA) was used to normalize membrane potential. The resting membrane potential was monitored for at least 5 min (basal), followed by the addition of pGH to the bath (5 μg/mL) for approximately 5 min. The frequency of action potentials was determined by analyzing the firing rate 2 min immediately prior to pGH administration and during the last 2 min of drug application. Changes in resting membrane potential were also evaluated in the presence of TTX (1 μM) and synaptic blockers (CNQX at 10 μM, AP-5 at 50 μM and picrotoxin at 50 μM). The membrane potential values were compensated to account for the junction potential (−8 mV).

**Evaluation of energy and glucose homeostasis.** The body weight of AgRP GHR KO, LepR GHR KO and brain GHR KO mice, as well as of their respective control animals, was monitored weekly. When mice reached approximately 20 weeks of age, they were single housed and food intake was daily measured for 5 to 7 consecutive days. Then, mice were subjected to a glucose tolerance test (2 g glucose/kg b.w.; i.p.) and to an insulin tolerance test (1 IU insulin/kg b.w.; i.p.). To

determine $O_2$ consumption (energy expenditure), $CO_2$ production, respiratory exchange ratio and locomotor activity (through infrared beam sensors), mice were placed in the Oxymax/Comprehensive Lab Animal Monitoring System (CLAMS; Columbus Instruments, Columbus, OH, USA). After an adaptation period of 3 days inside the CLAMS, these metabolic parameters were evaluated for 4 consecutive days. Therefore, the results presented were the average of this period. Total body fat and lean mass were measured by time-domain nuclear magnetic resonance (TD-NMR) using the LF50 body composition mice analyzer (Bruker, Germany). Body adiposity was also determined by summing the weight of the perigonadal, subcutaneous and retroperitoneal fat pads. The nose-anus length was assessed to determine body growth.

**Leptin and ghrelin responsiveness.** To assess leptin sensitivity, mice received an i.p. injection of either PBS or mouse recombinant leptin (2.5 μg/g b.w.; from Dr. A.F. Parlow, NHPP, USA) 3 h before their dark phase, and their food intake were recorded 4, 14 and 24 h following the injection. The food intake after PBS injection was compared with the food intake after leptin administration. The orexigenic response to ghrelin was determined in mice that received a subcutaneous injection of either PBS or ghrelin (0.2 μg/g b.w., Global Peptide, cat. no. C-et-004). Food intake was assessed 60 min after the injections.

**Metabolic effects induced by food restriction.** To investigate the neuroendocrine and metabolic changes induced by weight loss, mice were initially single housed and their food intake was recorded. Then, mice were subjected to a 60% food restriction protocol, in which each mouse received 40% of their normal intake 2 h before lights off for 5 to 7 consecutive days. During this period, the metabolic parameters were continuously assessed by the CLAMS, and their body weight, body composition (by TD-NMR) and glycemia were monitored at the time the food was provided. Daily calculation of $VO_2$ took into consideration the changes in body weight during food restriction to provide the value relative to body weight (ml/kg/h). Changes in oxygen consumption (energy expenditure) during food restriction were then reported as percentage of the values obtained from baseline (typically 2 to 3 days of recordings before food restriction). In addition, subgroups of adult (approximately 12-week-old) control and AgRP GHR KO mice were killed at the beginning of the light cycle (8:00 am) on the second day of food restriction. Mice with ad libitum access to food were killed at the same time. The hypothalamus, interscapular BAT and trunk blood were collected for subsequent analyses.

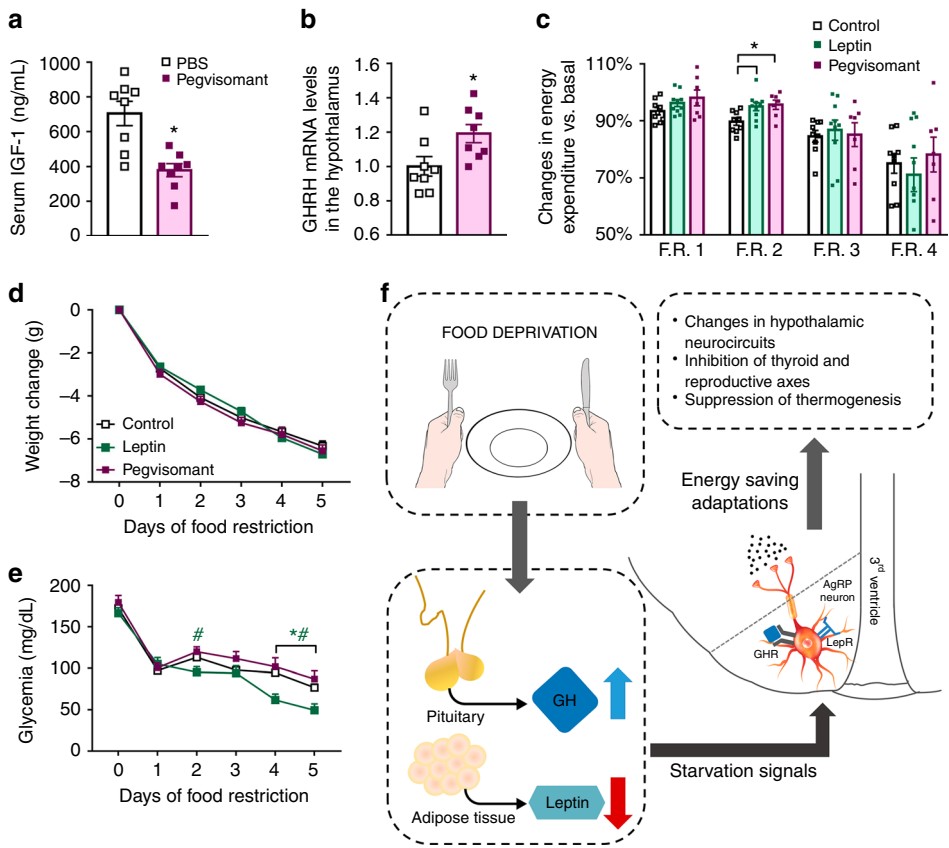

**Fig. 7** Pegvisomant produces energy-saving adaptations in food-deprived mice. **a** Serum insulin-like growth factor-1 (IGF-1) concentration ($t_{(14)} = 4.089$, $P = 0.0011$; $n = 8$). **b** Hypothalamic growth hormone-releasing hormone (GHRH mRNA levels ($t_{(14)} = 2.452$, $P = 0.0279$; $n = 8$). **c** Energy expenditure (food restriction (F.R.) 1, $F_{(2, 23)} = 1.83$, $P = 0.1829$; F.R. 2, $F_{(2, 23)} = 5.396$, $P = 0.012$; F.R. 3, $F_{(2, 23)} = 0.1271$, $P = 0.8813$; F.R. 4, $F_{(2, 23)} = 0.4397$, $P = 0.6496$; $n = 7–10$). **d** Body weight changes (main effect of F.R. [$F_{(5, 125)} = 1031$, $P < 0.0001$], main effect of treatment [$F_{(2, 25)} = 0.3717$, $P = 0.1163$] and interaction [$F_{(10, 125)} = 2.398$, $P = 0.0122$]; $n = 7–10$). **e** Blood glucose levels (main effect of F.R. [$F_{(5, 125)} = 116.6$, $P < 0.0001$], main effect of treatment [$F_{(2, 25)} = 4.747$, $P = 0.0179$] and interaction [$F_{(10, 125)} = 2.966$, $P = 0.0022$; $n = 8–10$; *$P < 0.05$, leptin vs. phosphate-buffered saline (PBS) treatment; #$P < 0.05$, leptin vs. pegvisomant treatment; Fisher's least significant difference (LSD) post-hoc test. **f** Scheme summarizing our findings highlighting that growth hormone (GH), parallel to the fall in leptin levels, is a critical cue that informs the brain about energy deficiency, triggering key adaptive responses to conserve body energy stores via activation of agouti-related protein (AgRP) neurons. All results were expressed as mean ± s.e.m.

**Drug treatment during food restriction**. Adult wild-type C57BL/6 male mice were subjected to the same 60% food restriction protocol described earlier, except that they received twice a day (at 9:00 am and 6:00 pm; lights on 8:00 am) i.p. injections of either mouse recombinant leptin (2.5 µg/g b.w. per injection; NHPP, USA), human GHR antagonist (pegvisomant; 20 µg/g b.w. per injection; Somavert®; Pfizer, Inc.) or control solution (pegvisomant diluent). Food deprivation-induced changes in energy expenditure and body weight were assessed as previously described.

**Metabolic effects induced by hypoglycemia**. To produce a counter-regulatory response to hypoglycemia, mice received an i.p. injection of 2DG (0.5 mg/kg b.w.; Sigma). We initially evaluated the effects of 2DG on glycemia for 180 min. Then, mice received i.p. injections of either PBS or 2DG and their food intake was recorded 2, 3 and 4 h afterwards.

**Hormone measurements**. Commercially available enzyme-linked immunosorbent assay (ELISA) kits were used to determine the serum concentration of leptin (Crystal Chem), T4 (Calbiotech), testosterone (Calbiotech), corticosterone (Arbor Assays), GH (Millipore), IGF-1 (R&D Systems) and prolactin (Sigma).

**Gene expression analysis**. Total RNA from the hypothalamus or interscapular BAT was extracted with TRIzol reagent (Invitrogen). Assessment of RNA quantity and quality was performed with an Epoch Microplate Spectrophotometer (Biotek). Total RNA was incubated in DNase I RNase-free (Roche Applied Science). Reverse transcription was performed with 2 µg of total RNA with SuperScript II Reverse Transcriptase (Invitrogen) and random primers p(dN)6 (Roche Applied Science). Real-time polymerase chain reaction was performed using the 7500TM Real-Time PCR System (Applied Biosystems) and Power SYBR Green PCR Master Mix

(Applied Biosystems). Relative quantification of mRNA was calculated by $2^{-\Delta\Delta Ct}$. Data were normalized to the geometric average of β-actin, glyceraldehyde 3-phosphate dehydrogenase (GAPDH) and cyclophilin A and reported as fold changes compared to values obtained from the control group (set at 1.0). The following primers were used: AgRP (forward: ctttggcggaggtgctagat; reverse: aggactc gtgcagccttacac), β-actin (forward: gctccggcatgtgcaaag; reverse: catcacacccttggtgcctc), cyclophilin A (forward: tatctgcactgccaagactgagt; reverse: cttcttgctggtcttgccattcc), GAPDH (forward: gggtcccagcttaggttcat; reverse: tacggccaaatccgttcaca), GHR (forward: atcaatccaagcctggggac; reverse: acagctgaatagatcctggggg), GHRH (forward: tat gcccggaaagtgatccag; reverse: atccttgggaatccctgcaaga), NPY (forward: cagatactactc cgctctcgcg; reverse: gggctggatctcttgccata), POMC (forward: tagatgtgtgggagctggtgc; reverse: ccagcgagaggtcgagtttg) and UCP-1 (forward: gaggtgtggcagtgttcattg; reverse: ggcttgcattctgaccttca).

**Statistical analysis**. All results were expressed as mean ± s.e.m. The paired two-tailed Student's $t$-test was used to compare the effects of GH or vehicle administration in the same animals, and in the electrophysiological data (before and during GH application). The unpaired two-tailed Student's $t$-test was used for comparisons between two groups. When three groups were compared simultaneously, we used one-way analysis of variance (ANOVA) and the Newman–Keuls multiple comparison tests. Data was analyzed using two-way ANOVA when appropriate, followed by Newman–Keuls or Fisher's least significant difference (LSD) post-hoc tests. Statistical analyses were performed using GraphPad Prism software. We considered $P$ values < 0.05 to be statistically significant.

**Reporting Summary**. Further information on experimental design is available in the Nature Research Reporting Summary linked to this article.

## Data availability
Source data for figures are available from the corresponding author upon request. A reporting summary for this Article is available as a Supplementary Information file.

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

## Acknowledgements
We thank Ana Maria P. Campos for expert technical assistance. J.D., R.F. and M.M. are financed by grants from the São Paulo Research Foundation (FAPESP; 17/02983–2, 17/21840–8 and 17/16473–6, respectively). I.C.F., G.O. and G.C. are financed by fellowships from FAPESP (16/09679–4, 17/05007–4 and 17/04006–4, respectively). We also thank the External Scholarship Program of the Argentine Research Council (CONICET) for supporting G.G.R.

## Author contributions
I.C.F. and J.D. conceived the project, designed the experiments, interpreted the results and wrote the manuscript. G.G.R. and M.P. performed the experiments with ghrelin. R.F. performed the electrophysiological experiments. I.C.F., P.D.S.T., G.O.S. and G.C.L.C. performed the remaining experiments. L.L.E., M.M., E.O.L. and J.J.K. provided essential equipment, reagents and expertise.

## Additional information

**Competing interests:** J.D. is recipient of the 2017 Global ASPIRE Young Investigator Research Awards in Endocrinology supported by Pfizer, Inc. The remaining authors declare no competing interests.

