## [Peer Review File · Nature Communications]

Reviewers' Comments:

Reviewer #1:

Remarks to the Author:

The paper by Furigo looks at an important area- the neuroendocrine adaptation to fasting and in particular how it may affect body weight.

The abstract makes quite a bold claims saying that GH signalling during food deprivation signals changes "...equivalent to those induced by declining leptin levels." Not quite so sure about the magnitude of the changes presented and there are several important areas that raise some concerns.

1. pharmacology vs physiology

a. The data in figure 1 looks to show that yes ,pharmacological doses of GH given ip activate agrp neurons and that given icv looks to increase food intake in mice . Not so sure about energy expenditure- is this still significant when analysed by ANCOVA with EE analysed against body weight, not as is currently written when corrected per KG?

b. Given this is a pharmacological experiment I think that the much more physiological statement in lines 75 and 76 is stated and needs to be reframed.

c. This is further borne out in the findings that agrp ghr KO mice looks to be identical to wild type mice – as line 90 states out , this perhaps could be a point to say Agrp GH receptor have little/if any role in day to day GH. This needs to be clearly stated.

2. Hormonal changes in fasting

a. How are we to know the (modest) changes in a- c occur in Agrp neurons? Has the c-fos been co-localised with agrp in these panels? Are you just assuming these are agrp neurons?

b. Why change from 24hr fasting in a- c to food restriction in panel d onwards? Much better surely to keep it clean with 24 hr fasting?

c. Line 104 " assessing if these hypothalamic changes affected thyroid, repro and adrenal axes"- not sure this is so and not certain if this direction is correct? Do not signals from the periphery signal TO the hypothalamus too.

d. I think essential to see serum Growth hormone in this figure panel (was it measured?) and pretty essential to see GH levels in the panel too in response to this partial food restriction? Move it out of the text.

e. Need to be clear in the text how long this food restriction is- 2 days for all? If so , why 2 days- surely much better to use 24 hr fasting to define the circuits?

f. Concerned that the significance in the corticosterone driven by 4 much higher results in control fasted when other groups are much lower- any comment here? Is this what might be driving the higher agrp in control fasted in panel e?

g. Looks too like t4 higher in the agrp ghr ko mice- why the discord with corticosterone?

h. Concerned that data in panel K appears to be labelled as significant. Are these not all paired to their baseline measurements? Should not be unpaired t test? those data on fr1 and fr2 look very similar. Also are these data, as ever, normalised to body weight (ancova analysis of EE vs body weight)

i. Would be good to have data in m and n as grams as well as relative

j. Why no change in leptin levels if a significant change in fat mass?

k. The data in panel O was more convincing as GH is known to rise a lot in hypoglycaemia and is likely to be a part of the hyperphagia of hypoglycaemia.

3. Loss of GHR from brain and leptin receptor positive cells

a. Panel 3e; No cre controls plotted; issues with nestin cre are well known; this should be here (see Harno et al cell metabolism 2013)

b. Panels 3 g and h – confusing , poorly labelled. One with 6 month old mice, another with 11 week old mice? I am confused.

c. The huge change in circulating changes in GH are odd- 1.6 vs 76.1 ??? indeed this whole section between lines 187 and 188 is odd and hard to follow? Data is presented in a rather random order, varying mice varying time point and is not compelling.

d. Previous comment on how to present energy expenditure apply to panel m- I cannot interpret this

4. Discussion

This is way too short, doesn't address many of the uncertainties, is not supported by the data and I think statements 187 and lines 203-206 are beyond the data in the paper. Needs a rewrite.

Reviewer #2:

Remarks to the Author:

This manuscript clearly shows novel and convincing evidences demonstrating, for the first time, that growth hormone acts in concert with leptin to alert the brain about energy deficiency, triggering key adaptive responses to conserve fuel stores. These results are significantly novel for the endocrinology and metabolism field since it is well-known that GH replacement is not sufficient to completely prevent the neuroendocrine adaptations induced by weight loss, indicating that additional, unknown, starvation factors are essential in this physiological process. The authors clearly demonstrate

novel and tremendously challenging animal models, that the central effects of GH are an additional key metabolic signal inducing, together with leptin, such adaptive responses and revealed the specific functional role of central GH signal in energy homeostasis in the whole-body organism. Therefore, the results included in this manuscript have a significant incremental novelty in the field (with original conclusions, work and statistical analysis appropriately conducted) and will definitely influence thinking in the field.

The reviewer has some questions:

- 1) The author uses porcine pituitary GH for i.p. administration in adult mice instead mouse GH. Please comment.
- 2) The authors indicated (line 104-105) that they assessed if the hypothalamic changes affected some hypothalamic-pituitary axes (thyroid, reproductive and adrenal) but they did not mention the prolactin axis? Did they measure PRL levels or other additional endpoints of this axis?
- 3) The authors logically mentioned the central role of leptin due to its well-known importance in this specific endpoint. Interestingly they did not find any difference in serum leptin between controls and AgRP GHR KO during food restriction (lines 109-111). Was this something that the authors expected? Had authors the opportunity to measure other key adipokines (adiponectin or resistin)?
- 4) The authors measure UCP-1 in AgRP GHR KO mice and show an increment of expression under food restriction compared to control. Did the authors measure UCP-1 levels in the BAT of the other KO mice model included in this study in order to determine if this interesting effect is also observed?
- 5) The animal model treated with pegvisomant (or leptin) should be further characterized in terms of GH axis and metabolic endpoints. Is pegvisomant in mice efficient to alter some of these parameters?
- 6) Why the authors only use male mice to perform these studies? There is a clear sexual dimorphism in the GH axis, therefore, this could be an important point to mention.

Reviewer #3:

Remarks to the Author:

This very well articulated and executed comprehensive study investigates the important function of GH and some of the neural populations it acts through to signal energy deficiency to the brain, imperative for initiating vital adaptive responses to conserve body energy stores. Overall, I am a proponent of this manuscript as the experiments are carefully designed and controlled for. Additionally, the data is presented in a clear manner and it is expertly written guiding the reader to the relevant data sets throughout. I have a few comments/inquiries/suggestions that in my opinion would significantly impact the findings that I have listed below (I have used an * to designate the most important points):

1. The authors demonstrate that i.c.v. admin of GH increases food intake over 24 hours but how rapid is this response? Does light cycle delivery of GH lead to acute food intake increases or is this more of a prolonged response that becomes apparent after 24 hours?
2. Recommend replacing "pharmacogenetic" with "chemogenetic" as the former means something else entirely (line 104).
3. The authors show that nearly all AgRP cells (91%) present STAT5 after GH admin but find only 25% respond to GH application in slice. How do the authors reconcile these large differences? Were the electrophysiology experiments done in the presence of drugs to isolate network activity?
4. * Figure 2a-c demonstrates that the levels of fasting induced Fos in AgRP neurons is reduced but this isn't specific to AgRP neurons. It would be far more convincing and supportive of their conclusions if this was done in a reporter background so this could be accurately determined. Moreover, having AgRP neurons marked by a fluorescent reporter like they do in Figure 1j would allow a nice acute brain slice experiment comparing the firing activity of AgRP neurons in control versus AgRP GHR KO. I would predict they would see an attenuation in both Fos activity and firing in AgRP neurons in the KO model.
5. * In a number of Figures (3a, S6b, S7a), it appears that the authors only quantified cell number and/or coexpression in a single coronal section. A more thorough analysis is required to make these conclusions. Counts should be made throughout the length of the entire arcuate nucleus (even if it is every 3rd or 4th section).
6. I may have missed it but did the authors report daily or weekly food intake measurements for the AgRP GHR KO mice compared to controls; this seems important.
7. Just curious if they have performed a fast-refeed experiment in these mice as I'd be interested to see if they have blunted a response in this condition.
8. * The authors demonstrate differential effects with LepR and whole brain GHR KO compared to both each other and controls. It would be insightful to address potential cell types or anatomical regions that may be responsible for these differential effects.

the Discussion.

Print Email

Resend E-mail

RESPONSE TO REVIEWERS

Reviewer #1:

The paper by Furigo looks at an important area- the neuroendocrine adaptation to fasting and in particular how it might affect body weight.

The abstract makes quite a bold claims saying that GH signalling during food deprivation signals changes “....equivalent to those induced by declining leptin levels.” Not quite so sure about the magnitude of the changes presented and there are several important areas that raise some concerns.

1. pharmacology vs physiology

a. The data in figure 1 looks to show that yes ,pharmacological doses of GH given ip activate agrp neurons and that doses given icv looks to increase food intake in mice . Not so sure about energy expenditure- is this still significant when analysis by ANCOVA with EE analysed against body weight, not as is currently written when corrected per KG?

We would like to thank the reviewer for the valuable comments and suggestions. Regarding the energy expenditure results after icv GH infusion, we reanalyzed the data without correcting for body weight (ml/hr instead of ml/kg/hr) and VO_2 was still reduced after central GH injection (see figure below; $t_{(5)} = 3.039$, $P = 0.0288$, paired t test). Thus, we decided to keep the original figure (data corrected for kg) in the revised manuscript.

b. Given this is a pharmacological experiment I think that the much more physiological statement in lines 75 and 76 is over stated and needs to be reframed.

The referred sentence was rephrased as “*these findings indicate that exogenous administration of GH induces an orexigenic response via activation of AgRP neurons.*”

c. This is further borne out in the findings that agrp ghr kO mice looks to be identical to wild type mice – as line 90 points out , this perhaps could be a point to say Agrp GH receptor have little/if any role in day to day GH. This needs to be more clearly stated.

The phrase was revised according to the reviewer’s comment: “*These results suggest that GHR expression in AgRP neurons is unnecessary for the regulation of energy homeostasis under normal circumstances or for the response to key hormones that rely on AgRP neurons to modulate energy homeostasis. Thus, endogenous fluctuations of plasma GH levels likely do not play an important role modulating the energy balance in ad libitum fed conditions.*”

2. Hormonal changes in fasting

a. How are we to know the (modest) changes in a- c occur in Agrp neurons? Has the c-fos been co-localised with agrp in these panels? Are you just assuming these are agrp neurons?

In order to specifically determine the activity of AgRP neurons, we measured the number of fasting-induced c-Fos positive cells in brain series of AgRP-Cre/tdTomato-reporter (control) mice and in AgRP-Cre/GHR^{flox/flox}/tdTomato-reporter (AgRP GHR KO) mice. We observed that the total number of c-Fos positive cells in the ARH was reduced in AgRP GHR KO mice. The co-localization showed that this reduction was explained by a lower number of AgRP cells positive for c-Fos, while the number of non-AgRP cells positive

for c-Fos remained unchanged between the groups. These co-localization data were now added in the revised manuscript (Figures 2a-c).

b. Why change from 24hr fasting in a- c to food restriction in panel d onwards? Much better surely to keep it clean and stay with 24 hr fasting?

The c-Fos experiment was performed after 24 hr fasting. The remaining experiments were performed in mice that received an amount of food that represented 40% of their usual intake. Thus, we believe that food restriction represents better the metabolic condition of the animals. In addition, a detailed explanation of why we used food restriction rather than fasting was mentioned in the following responses (please, see the response to item e; page 4).

C. Line 104 “ assessing if these hypothalamic changes affected thyroid, repro and adrenal axes”- not sure this is so linear and not certain if this direction is correct? Do not signals from the periphery signal TO the hypothalamus too.

The statement was rephrased as “*Next, we assessed whether AgRP GHR KO mice exhibit a normal endocrine response to weight loss*”.

d. I think essential to see serum Growth hormone in this figure panel (was it measured?) and pretty essential to see leptin levels in the panel too in response to this partial food restriction? Move it out of the text.

As suggested by the reviewer, the data showing leptin and GH circulating levels was added as Figure 2j and Figure 2k, respectively. Briefly, we showed that while food restriction reduced serum leptin levels and increased serum GH concentration, GHR ablation in AgRP cells caused no significant effects in these responses.

e. Need to be clear in the text how long this food restriction is- 2 days for all? If so , why 2 days- surely much better to do 24 hr fasting to define the circuits?

In these groups, all tissue collection was performed after 2 days of food restriction (40% of the normal intake). This information was also added in the Results section. While our group and others have previous experience using 24 hours of fasting to induce c-Fos expression in the ARH (Liu *et al.*, *Neuron* 73:511-522, 2012; Pedroso *et al.*, *Endocrinology* 157:3901-3914, 2016), we decided to use a food restriction protocol that had been shown to increase GH secretion in order to study the neuroendocrine effects of GH signaling in AgRP neurons (Zhao *et al.*, *Proc Natl Acad Sci U S A* 107:7467-7472, 2010; Li *et al.*, *J Biol Chem* 287:17942-17950, 2012). Of note, our food restriction protocol produced similar neuroendocrine effects than those caused by fasting in previous studies, including suppression of thyroid and reproductive axes, increased glucocorticoid secretion, increase in hypothalamic AgRP and NPY expression and decreased POMC expression (Ahima *et al.*, *Nature* 382:250-252, 1996; Pedroso *et al.*, *Endocrinology* 157:3901-3914, 2016). Part of this explanation was included in the Discussion section of the revised manuscript.

f. Concerned that the significance in the corticosterone driven by 4 much higher results in control fasted when other mice much much lower- any comment here? Is this what might be driving the higher agrp in control fasted in panel e?

We believe that these results were caused by individual variations of each animal and the two-way ANOVA revealed a significant effect of food restriction [$F_{(1, 49)} = 16.13$, $P = 0.0002$] and an interaction with GHR ablation [$F_{(1, 49)} = 4.13$, $P = 0.0476$]), which could be interpreted that the increase in corticosterone levels induced by food restriction is influenced by GHR ablation in AgRP cells. In addition, a linear correlation between

hypothalamic AgRP mRNA levels and serum corticosterone concentration did not show significant *P* value in AgRP GHR KO mice during food restriction.

g. Looks too like t4 higher in the agrp ghr ko mice- why the discord with corticosterone?

AgRP GHR KO mice showed similar T4 concentration in *ad libitum*-fed mice compared to control animals, although AgRP GHR KO group showed a higher variability than the control group. In the PVH, AgRP signaling activates CRH neurons and inhibits TRH neurons (Fekete et al., 2002; Dimitrov et al., 2007). Wild-type mice under food restriction display increased AgRP expression (Fig. 2e), so an increase in corticosterone and a decrease of T4 is expected. AgRP GHR KO mice displayed a smaller increase in AgRP expression; thus, less evident increments in corticosterone and a smaller decrease of T4 is expected.

h. Concerned that data in panel K appears to be labelled as significant. Are these not all paired to their baseline measure so should not be unpaired t test? those data on fr1 and fr2 look very similar. Also are these data, as ever, normalised to change in weight (ancova analysis of EE vs body weight)

These data were now moved to Figure 3a. The differences between groups were assessed by unpaired t test. We also reduced the size of symbols representing individual values to facilitate the visualization of means and the differences between groups. The original data were normalized by body weight (ml/kg/hr), and then we showed the results as the percentage of reduction compared to baseline (represented as 100%). According to the reviewer suggestion, the effect of weight change in each day of food restriction was separated from the residue using multiple linear regression, while the VO₂ values were computed without kg correction. Now, differences were observed for all days of food restriction,

including F.R. 4. A summary of this analysis is presented below (each table represents each day of food restriction):

Regression Summary for Dependent Variable: RA1 (ancova 2018 in ancova 2018)						
R= ,64691058 R ² = ,41849329 Adjusted R ² = ,36034262						
F(2,20)=7,1967 p<,00442 Std.Error of estimate: 4,1159						
N=23	b*	Std.Err. of b*	b	Std.Err. of b	t(20)	p-value
Intercept			85,99066	5,784223	14,86641	0,000000
GRUPO	0,586590	0,173312	5,91032	1,746243	3,38459	0,002944
? RA1	0,397231	0,173312	5,21131	2,273698	2,29200	0,032888

Regression Summary for Dependent Variable: RA2 (ancova 2018 in ancova 2018)						
R= ,52963959 R ² = ,28051810 Adjusted R ² = ,20856991						
F(2,20)=3,8989 p<,03717 Std.Error of estimate: 5,2149						
N=23	b*	Std.Err. of b*	b	Std.Err. of b	t(20)	p-value
Intercept			72,88088	10,33988	7,048523	0,000001
GRUPO	0,532085	0,190878	6,10668	2,19068	2,787567	0,011365
? RA2	0,091117	0,190878	1,38904	2,90985	0,477357	0,638283

Regression Summary for Dependent Variable: RA3 (ancova 2018 in ancova 2018)						
R= ,54010981 R ² = ,29171861 Adjusted R ² = ,22089047						
F(2,20)=4,1187 p<,03177 Std.Error of estimate: 5,1324						
N=23	b*	Std.Err. of b*	b	Std.Err. of b	t(20)	p-value
Intercept			65,02643	10,90077	5,965306	0,000008
GRUPO	0,534786	0,196620	6,08820	2,23840	2,719895	0,013190
? RA3	-0,017481	0,196620	-0,23212	2,61079	-0,088908	0,930040

Regression Summary for Dependent Variable: RA4 (ancova 2018 in ancova 2018)						
R= ,59906511 R ² = ,35887900 Adjusted R ² = ,29476690						
F(2,20)=5,5977 p<,01173 Std.Error of estimate: 5,0219						
N=23	b*	Std.Err. of b*	b	Std.Err. of b	t(20)	p-value
Intercept			83,32057	10,13638	8,219956	0,000000
GRUPO	0,554178	0,184767	6,48847	2,16330	2,999337	0,007086
? RA4	0,402402	0,184767	4,49411	2,06352	2,177890	0,041556

Based on this ANCOVA analysis, we decided to reanalyze all our VO₂ data during food restriction and take into account the changes in body weight during food restriction in the daily calculation of VO₂ (instead of just using the initial body weight for normalization when the mouse was placed in the CLAMS). Using this calculation methodology, we basically found the same results than previously observed with only a few exceptions (e.g., a

difference in F.R. 4 in AgRP GHR KO mice as shown by the ANCOVA). Thus, in the revised manuscript, we updated the VO₂ data during food restriction of AgRP GHR KO model, LepR GHR KO model, Brain GHR KO model and during leptin or pegvisomant treatment.

i. Would be good to have data in m and n as grams as well as relative

The results in grams are exhibited below and basically they show a similar data as those presented in the normalized results (GHR ablation in AgRP cells affects the degree of reduction in body weight). Since the manuscript already has a large number of figures, including 16 supplementary figures, we choose to show only the relative results in the manuscript.

j. Why no change in leptin levels if a significant change in fat mass?

Food restriction induces a very robust fall in leptin levels and this reduction usually occurs at a greater magnitude than the loss of body fat. Thus, during this situation, leptin levels may not reflect precisely body adiposity, explaining why AgRP GHR KO mice showed no change in serum leptin levels, while they lost more fat mass during food restriction, compared to control animal. We included a sentence in the revised manuscript discussing this possibility. The sentence states as follows: “F.R. induces a very robust fall in leptin levels

and this reduction usually occurs at a greater magnitude than the loss of body fat^{3,35,36}. Therefore, during this situation, leptin levels may not reflect precisely body adiposity, explaining why AgRP GHR KO mice showed no change in serum leptin levels, while they lost more fat mass during F.R. compared to control animal.”

k. The data in panel O was more convincing as GH is known to rise a lot in hypoglycaemia and is likely to be a partial driver of the hyperphagia of hypoglycaemia.

We thank the reviewer for this observation. These data were now moved to Figure 3f. As pointed out by the reviewer, GH likely plays a role during hypoglycaemia since GH levels marked increase during this condition and GH can cause insulin resistance and increase blood glucose levels. Our findings also suggest that this increase in GH secretion may activate AgRP neurons favoring the typical hypoglycaemia-induced hyperphagia.

3. Loss of GHR from brain and leptin receptor positive cells

a. Panel 3e; No cre controls plotted; issues with nestin cre are well known; this should be here (see Harno et al cell metabolism 2013)

We are aware of the phenotype that can be caused by the Nestin-cre transgene expression. Our major model was the AgRP-Cre and we used the Nestin-Cre (and the LepR-Cre as well) to confirm some of the phenotypes induced by GHR ablation in AgRP cells. However, a key aspect of the Nestin-Cre is a small but ectopic expression of GH in neurons, which leads to reduction in pituitary GH secretion (secondary to the activation of negative feedback loops in the hypothalamus) and consequently impaired growth and metabolic abnormalities (Harno *et al.*, *Cell Metab* 18:21-28, 2013; Declercq *et al.*, *PLoS One* 10:e0135502, 2015). Interestingly, our conditional knockout mouse pretty much “corrects” the defects of Nestin-Cre mouse, since the deletion of GHR prevents the effects of the central

GH production from interfering with pituitary production of this hormone. It is no wonder that brain GHR KO mice show increased growth, instead of reduced somatic growth, which is the major phenotype of Nestin-Cre transgenic mouse. The Discussion section of the new version of the manuscript acknowledges this fact as follows: “*In addition, Nestin-Cre transgene expression was previously shown to produce a phenotype per se which could have interfered with the responses observed in the present study*^{41,42}”

b. Panels 3 g and h – confusing , poorly labelled. One with 6 month old mice, another with 11 week old mice? I am confused

At the time of the experiment, we had just received the NMR equipment for body composition analysis and we only had available for evaluation animals of different ages. In the revised manuscript, all body composition analyses were performed in 6 month old mice. The updated data are available in Figure 4 and we also improved the labelling of the figure.

c. The huge change in circulating changes in GH are odd- 1.6 vs 76.1 ??? indeed this whole section between lines 137 to 155 is odd and hard to follow? Data is presented in a rather random order, varying mice varying time point and is not compelling.

Based on the reviewer’s comment, we decided to remove these results from the revised manuscript.

d. Previous comment on how to present energy expenditure apply to panel m- I cannot interpret this

As previously mentioned, the VO₂ data during food restriction were reanalyzed in the revised manuscript.

4. Discussion

This is way too short, doesn't address many of the uncertainties, is not supported by the data and I think statement in lines 187 and lines 203-206 are beyond the data in the paper.

Needs a rewrite.

The Discussion section was expanded in the revised manuscript and reformulated according to the reviewer's comments.

Reviewer #2:

This manuscript clearly shows novel and convincing evidences demonstrating, for the first time, that growth hormone (GH) acts in concert with leptin to alert the brain about energy deficiency, triggering key adaptive responses to conserve limited fuel stores. These results are significantly novel for the endocrinology and metabolism field since it is well-known that leptin replacement is not sufficient to completely prevent the neuroendocrine adaptations induced by weight loss, indicating that additional, unknown, starvation factors are essential in this physiological process. The authors clearly demonstrate, using novel and tremendously challenging animal models, that the central effects of GH are an additional key metabolic signal inducing, together with leptin, such adaptative responses and revealed the specific functional role of central GH signaling for energy homeostasis in the whole-body organism. Therefore, the results included in this manuscript have a significant and incremental novelty in the field (with original conclusions, work and statistical analysis appropriately conducted) and, definitely, will influence thinking in the field.

The reviewer has some questions:

1) The author uses porcine pituitary GH for i.p. administration in adult mice instead mouse GH. Please comment.

We would like to thank the reviewer for his/her comments and suggestions. We have experience using either mouse GH (mGH) or porcine (pGH) to induce STAT5 phosphorylation in the mouse brain, and both are able to induce pSTAT5 in the same brain areas. We also consulted Dr. A.F. Parlow from the National Hormone and Peptide Program (July 2006), who is a specialist in the production and action of pituitary hormones. He recommended the use of pGH in our experiments due to the great similarity to mGH and rat GH. As shown in our validation studies, pGH produces a very specific pSTAT5 staining,

which is absent in ablated cells of our tissue-specific knockout mice. Therefore, pGH seems to be a very good GHR agonist in mouse tissues.

2) The authors indicated (line 104-105) that they assessed if the hypothalamic changes affected some hypothalamic-pituitary axes (thyroid, reproductive and adrenal) but they did not mention the prolactin axis? Did they measured circulating PRL levels or other additional endpoints of this axis?

As suggested by the reviewer, we now analyzed circulating prolactin levels in AgRP GHR KO mice and their respective control animals, but no significant effect was observed either for food restriction or GHR ablation. This result was added in the revised manuscript as supplementary figure 8. We also assessed hypothalamic TH mRNA expression during food restriction and in *ad libitum* fed mice and no significant differences among the groups were observed (data now shown).

3) The authors logically mentioned the central role of leptin due to its well-known importance in this specific endpoint. Interestingly they did not find any difference in serum leptin between controls and AgRP GHR KO during food restriction (lines 109-111). Was this something that the authors expected? Had authors the opportunity to measure other key adipokines (adiponectin or resistin)?

Food restriction induces a very robust fall in leptin levels and this reduction usually occurs at a greater magnitude than the loss of body fat. Thus, during this situation, leptin levels may not reflect precisely body adiposity, explaining why AgRP GHR KO mice showed no change in serum leptin levels, while they lost more fat mass during food restriction, compared to control animal. We included a sentence in the revised manuscript discussing this

possibility. We did not measure the levels of other adipokines since these analyses were out of the scope of our study.

4) The authors measure UCP-1 in AgRP GHR KO mice and show an increment of expression under food restriction compared to control. Did the authors measure UCP-1 levels in the BAT of the other KO mice model included in this study in order to determine if this interesting effect is also observed?

We also measured UCP-1 expression in the BAT of LepR GHR KO mice and we observed a similar effect than that observed in AgRP GHR KO mice (Figure 4L). Briefly, while food restriction suppressed UCP-1 expression in the BAT of control mice, GHR ablation in LepR-expressing cells prevented this reduction.

5) The animal model treated with pegvisomant (or leptin) should be further characterize in terms of GH axis and metabolic endpoints. Is pegvisomant in mice efficient to alter some of these parameters?

As suggested by the reviewer, we further characterized whether pegvisomant can affect the GH axis in mice. Thus, C57BL/6 mice received twice a day i.p. injections of pegvisomant or PBS (similar to the experiment shown in the initial submission) and on the fourth day of treatment we collected blood samples and their hypothalami. We observed that pegvisomant treatment reduced circulating IGF-1 levels and induced an upregulation of GHRH mRNA in the hypothalamus. These results clearly indicate impairment in GH signaling in pegvisomant treated mice, demonstrating that this drug acts as a GHR antagonist in mice, as predicted in our initial experiments. These results were added in the revised manuscript (Figures 5a-b), as well as a paragraph discussing them.

6) Why the authors only use male mice to perform these studies? There is a clear sexual dimorphism in in the GH axis and therefore, this could be a important point to mention

In the revised manuscript, we included data regarding AgRP GHR KO females (supplementary figure 11). As seen in males, AgRP GHR KO females showed an attenuated increase in hypothalamic AgRP and NPY mRNA levels during food restriction, whereas POMC expression was not affected by GHR ablation. However, we could not observe a decrease in energy expenditure of AgRP GHR KO females during food restriction, compared to control animals. Despite of that, AgRP GHR KO females exhibited an increased weight loss and decreased glycemia during food restriction, which is a similar result to that observed in males. In addition, we tried to assess circulating estradiol levels during food restriction and in *ad libitum* fed mice using a Mouse/Rat Estradiol ELISA kit (Calbiotech). However, most of the samples were below the detection limit and we were unable to use these data (data not shown).

Reviewer #3:

*This very well articulated and executed comprehensive study investigates the important function of GH and some of the neural populations it acts through to signal energy deficiency to the brain, imperative for initiating vital adaptive responses to conserve body energy stores. Overall, I am a proponent of this manuscript as the experiments are carefully designed and controlled for. Additionally, the data is presented in a clear manner and it is expertly written guiding the reader to the relevant data sets throughout. I have a few comments/inquiries/suggestions that in my opinion would significantly enhance the findings that I have listed below (I have used an * to designate the most important points):*

1. The authors demonstrate that icv admin of GH increases food intake over 24 hours but how rapid is this response? Does light cycle delivery of GH lead to acute food intake increases or is this more of a prolonged response that becomes obvious after 24 hours?

We would like to thank the reviewer for the valuable comments and suggestions. An acute icv GH injection did not affect the food intake in the following 30, 60, 120 and 240 minutes. Thus, GH effect on food intake is not as fast as that observed after ghrelin or leptin administration. These data were added as Figure 1d.

2. Recommend replacing "pharmacogenetic" with "chemogenetic" as the former means something else entirely (line 70).

As suggested, the term "pharmacogenetic" was replaced.

3. The authors show that nearly all AgRP cells (91%) present STAT5 after GH admin but find only 25% respond to GH application in slice. How do the authors reconcile these large differences? Were the ephys experiments done in the presence of drugs to isolate network activity?

Regarding the differences in the percentage of cells responsive to GH using pSTAT5 or in the ephys experiments, we believe that this is a common characteristic of cytokine receptors. For example, several studies investigated the effects of leptin in the electrical activity of POMC and AgRP neurons. Although nearly 80% of POMC or AgRP neurons express the leptin receptor (Baquero *et al.*, *J Neurosci* 34:9982-9994, 2014; Lima *et al.*, *Brain Res* 1646:366-376, 2016), leptin induces a depolarization between 20% and 66% of ARH POMC cells, depending of the rostrocaudal level, and inhibits ~ 57% of ARH AgRP neurons (Williams *et al.*, *J Neurosci* 30:2472-2479, 2010; Baquero *et al.*, *J Neurosci* 34:9982-9994, 2014). Thus, not all neurons that express leptin receptor show electrical changes to leptin application, even in well-known areas that are responsive to this hormone. Perhaps, the same occurs with GHR, even in a greater magnitude, which could indicate that the genomic actions of GH (probably via STAT5 transcription factors) are more important than its acute effects on resting membrane potential. We included a sentence in the revised manuscript discussing this idea. As suggested by the reviewer, we measured the effects of GH using voltage-gated sodium channel antagonist TTX and synaptic blockers in order to determine whether the effects of GH are direct in AgRP cells. Similar to the results shown earlier, GH application in the presence of TTX and synaptic blockers depolarized 25% of ARH AgRP neurons (3 out 12 recorded cells from 4 mice), changing in $+7.7 \pm 1.4$ mV their resting membrane potential ($t_{(2)} = 5.277, P = 0.0341$). This new result was added in the revised manuscript.

4. * Figure 2a-c demonstrates that the levels of fasting induced Fos in is reduced but this isn't specific to AgRP neurons. It would be far more convincing and supportive of their conclusions if this was done in a reporter background so this can be accurately determined. Moreover, having AgRP neurons marked by a fluorescent reporter like they do in Figure 1j would allow a nice acute brain slice experiment comparing the firing activity

of AgRP neurons in control versus AgRP GHR KO mice. I would predict they would see an attenuation in both Fos activity and firing in AgRP neurons in the KO model.

In order to specifically determine the activity of AgRP neurons, we measured the number of fasting-induced c-Fos positive cells in brain series of AgRP-Cre/tdTomato-reporter (control) mice and in AgRP-Cre/GHR^{flox/flox}/tdTomato-reporter (AgRP GHR KO) mice. We observed that the total number of c-Fos positive cells in the ARH was reduced in AgRP GHR KO mice. The co-localization showed that this reduction was explained by a lower number of AgRP cells expressing c-Fos, while the number of non-AgRP cells expressing c-Fos remained unchanged between the groups. The co-localization data were now added in the revised manuscript (Figures 2a-c). Regarding recording the activity of AgRP neurons using brain slices of AgRP GHR KO mice carrying a reporter protein, unfortunately we cannot perform this experiment at this time because these mice are not available.

5. * In a number of Figures (3a, S6b, S7a), it appears that the authors only quantified cell number and/or coexpression in a single coronal section. A more thorough analysis is required to make these conclusions. Counts should be made throughout the length of the entire arcuate nucleus (even if it is every 3rd or 4th section).

In all histological experiments, we counted the number of cells in two or three sections (rostrocaudal levels) of each nucleus. The results of each animal represent mean values. In addition, the rostrocaudal extension of each quantified area was previously analyzed to make sure that the observed pattern was similarly distributed. In all brain areas analyzed, we did not observe changes between different rostrocaudal levels.

6. *I may have missed it but did the authors report daily or weekly food intake measurements for the AgRP GHR KO mice compared to controls; this seems important.*

Food intake was daily measured for 5 to 7 consecutive days and we presented the average of these measures.

7. *Just curious if they have performed a fast-refeed experiment in these mice as I'd be interested to see if they have a blunted a response in this condition.*

We assessed the food intake of a subgroup of mice during refeeding, but no differences between the groups were observed (see figure below). Since it was not the scope of our study to investigate the refeeding, this result was not included in the manuscript.

8. ** The authors demonstrate differential effects with LepR and whole brain GHR KO compared to both each other and AgRP. It would be insightful to address potential cell types or anatomical regions that may be responsible for these differences in the Discussion.*

The Discussion section was expanded in the revised manuscript and we included a paragraph (page 12) discussing potential neuronal populations responsible for the differences observed among the mouse models.

Reviewers' Comments:

Reviewer #1:

Remarks to the Author:

This is improved and on the whole a much better looking manuscript from my perspective.

Looking at the response to my comment as reviewer #1, majority answered fine.

a few small things

Point 1a. fine , glad it still holds up with this analysis .i suggest swapping out original for the graph in the rebuttal.

Point 2 j. sorry, still confused. Aware that with fasting , circulating leptin levels falls away before a clear demonstrable falling away of fat mass but here there was an animal that lost fat mass and yet still did not have a change in leptin levels. Your sentence doesn't make sense or follow logic. If you don't know what this happens, could just say so and maybe speculate on changes in autonomic innervation or changes in hormonal milieu in the difference animals.

A few small point

Abstract line 25- "similar to those induced by declining leptin"- not true- take this out the sentence and stop after "during food deprivation".

Line 54- "unveiled yet" – bit awkward try "has not been fully defined"

Line 68- clarify the time here. Nothing seems ot happen at several of these time point but just a difference at 24hr?

Line 224- "super imposing" – disagree- should say GH and leptin both play a role informing.."

Line 229 "therefore the brain... not helpful , remove this line.

Line 252- see comment on point 2.j. above this doesn't make sense. Not a deal breaker but what you have said needs to be rewritten. Happy if this is left speculative

Figs 1 d- are the labels the right way round? Looks like GH makes mice eat less over the first 4 hrs

Fig 3b – are there no error bars here? Also mice number 5-9 ; only 2 groups; just say which group had 5 which had 9.

Legends these are really dense now with all the stats in them and are very hard to read. Look to try and make this clearer- also look to make sure that the letter or panels are present fig 1 d – e, why is e in the wrong place? Fig 2 g-l, horrible to read now in a big block of text , need g, text, h, text, l, text, l, text

Reviewer #2:

Remarks to the Author:

Authors have satisfactorily answered all my previous concerns/questions and have nicely revised the manuscript which I consider that include novel and convincing evidences demonstrating that GH acts in concert with leptin to alert the brain about energy deficiency, triggering key adaptive responses to conserve limited fuel stores. Therefore, the data included in this new version of the manuscript have a significant and incremental novelty in the field (with original conclusions, work and statistical analysis appropriately conducted, novel and tremendously challenging animal models, etc.) and, definitely, will influence thinking in the endocrinology and metabolism field.

Reviewer #3:

Remarks to the Author:

The authors have adequately addressed my concerns.

RESPONSE TO REVIEWERS

Reviewer #1:

This is improved and on the whole a much better looking manuscript from my perspective.

Looking at the response to my comment as reviewer #1, majority answered fine.

a few small things

Point 1a. fine , glad it still holds up with this analysis .i suggest swapping out original for the graph in the rebuttal.

We would like to thank the reviewer for the valuable comments and suggestions. The Figure 1e was replaced as suggested by the reviewer.

Point 2 j. sorry, still confused. Aware that with fasting , circulating leptin levels falls away before a clear demonstrable falling away of fat mass but here there was an animal that lost fat mass and yet still did not have a change in leptin levels. Your sentence doesn't make sense or follow logic. If you don't know what this happens, could just say so and maybe speculate on changes in autonomic innervation or changes in hormonal milieu in the difference animals.

As suggested by the reviewer, we reformulated the discussion about the lack of changes in serum leptin levels during food restriction as follows:

“Additionally, AgRP GHR KO mice showed no change in serum leptin levels compared to control animal, while they lost more fat mass during F.R. Although the reasons for the lack of difference in circulating leptin levels in animals with different degrees of adiposity are unknown, it is possible to speculate that changes in autonomic innervation or hormonal milieu could have affected leptin synthesis and secretion by adipose tissue. In fact, POMC and AgRP neurons modulate autonomic nerve activity in various tissues, including fat

depots³⁴, and these neurons can regulate fasting-induced fall in leptin levels, independently of changes in fat mass³⁵.”

A few small point

Abstract line 25- “similar to those induced by declining leptin”- not true- take this out the sentence and stop after “during food deprivation”.

The abstract was changed accordingly.

Line 54- “unveiled yet” – bit awkward try ”has not been fully defined”

The sentence was changed according to reviewer’s suggestion.

Line 68- clarify the time here. Nothing seems ot happen at several of these time point but just a difference at 24hr?

The sentence was modified as follows:

“Although icv administration of GH caused no significant changes in food intake during the first 4 hours of measurement, C57BL/6 mice exhibited increased food intake and reduced energy expenditure 24 hours after the injection (Fig. 1d-e).”

Line 224- “super imposing” – disagree- should say GH and leptin both play a role informing..”

The sentence was changed according to reviewer’s suggestion.

Line 229 “therefore the brain... not helpful , remove this line.

As suggested, this phrase was removed from the revised manuscript.

Line 252- see comment on point 2.j. above this doesn't make sense. Not a deal breaker but what you have said needs to be rewritten. Happy if this is left speculative

As mentioned in an earlier response, the sentence that discusses serum leptin levels during food restriction was modified according to reviewer's comments.

Figs 1 d- are the labels the right way round? Looks like GH makes mice eat less over the first 4 hrs

Data were shown correctly and there was no significant difference in the first 4 hours.

Fig 3b – are there no error bars here? Also mice number 5-9 ; only 2 groups; just say which group had 5 which had 9.

There are error bars in the graph, although they are very small. The sample size in this experiment (Fig. 3b) was: control = 25 and AgRP GHR KO = 22. The long figure legend probably impaired the reading. The figure legends were revised and the sample size of each group was defined.

Legends these are really dense now with all the stats in them and are very hard to read. Look to try and make this clearer- also look to make sure that the letter or panels are present fig 1 d – e, why is e in the wrong place? Fig 2 g-l, horrible to read now in a big block of text , need g, text, h, text, I, text, l, text

The figure legends were revised. The previous Figures 2 and 4 were split into two figures each, in order to meet the maximum word limit of each legend (350 words). Now the revised manuscript has 7 Figures.

Reviewer #2:

Authors have satisfactorily answered all my previous concerns/questions and have nicely revised the manuscript which I consider that include novel and convincing evidences demonstrating that GH acts in concert with leptin to alert the brain about energy deficiency, triggering key adaptive responses to conserve limited fuel stores. Therefore, the data included in this new version of the manuscript have a significant and incremental novelty in the field (with original conclusions, work and statistical analysis appropriately conducted, novel and tremendously challenging animal models, etc.) and, definitely, will influence thinking in the endocrinology and metabolism field.

We would like to thank the reviewer for his/her comments.

Reviewer #3:

The authors have adequately addressed my concerns.

We would like to thank the reviewer for his/her comments.